# Randomized Quantization is All You Need for Differential Privacy in Federated Learning

## Abstract

Federated learning (FL) is a common and practical framework for learning a machine model in a decentralized fashion. A primary motivation behind this decentralized approach is data privacy, ensuring that the learner never sees the data of each local source itself. Federated learning then comes with two majors challenges: one is handling potentially complex model updates between a server and a large number of data sources; the other is that de-centralization may, in fact, be insufficient for privacy, as the local updates themselves can reveal information about the sources' data. To address these issues, we consider an approach to federated learning that combines quantization and differential privacy. Absent privacy, Federated Learning often relies on quantization to reduce communication complexity. We build upon this approach and develop a new algorithm called the **R**andomized **Q**uantization **M**echanism (RQM), which obtains privacy through a two-levels of randomization. More precisely, we randomly sub-sample feasible quantization levels, then employ a randomized rounding procedure using these sub-sampled discrete levels. We are able to establish that our results preserve "Rényi differential privacy" (Rényi DP). We empirically study the performance of our algorithm and demonstrate that compared to previous work it yields improved privacy-accuracy trade-offs for DP federated learning. To the best of our knowledge, this is the first study that solely relies on randomized quantization without incorporating explicit discrete noise to achieve Rényi DP guarantees in Federated Learning systems.

## 1 Introduction

Federated Learning (FL) is an innovative approach to training on massive datasets, utilizing a multitude of devices like smartphones and IoT devices, each containing locally stored, privacy-sensitive data. At a basic level, privacy is maintained by storing local data on each end-user device without sharing it with the server. However, in some cases, device or local device data can be partially reconstructed from computed gradients (Zhu et al., 2019). This potential data leakage from gradients can be addressed through the use of privacy-preserving techniques. Equally important in the context of FL is communication efficiency. Given the extensive communication demands placed on many edge computing devices and the constraints of limited bandwidth, it is imperative to devise a training scheme that not only preserves privacy but also aligns with the requirements of efficient communication within FL systems. What is perhaps surprising, however, is that these two objectives are not necessarily in tension and *can even be aligned!* One way to improve communication overhead is to reduce bit complexity through stochastic rounding schemes, but we show that these randomization procedures, if designed carefully, provide additional benefits to data privacy.

Past studies, such as the work of Agarwal et al. (2018); Kairouz et al. (2021); Agarwal et al. (2021), have sought to tackle this issue by employing various forms of discrete additive DP noise in conjunction with quantization; this is in part because quantization has immediate communication complexity benefits over continuous noise. However, when these discrete additive noise methods are coupled with secure aggregation protocols (Bonawitz et al., 2017), aimed at preventing a server from inspecting individual local device updates, they encounter a challenge of biased estimation due to modular clipping. To solve this, Chen et al. (2022) introduce the Poisson Binomial Mechanism (PBM), bypassing the use of additive noise and instead directly mapping continuous inputs to discrete values in an unbiased fashion.

Losses in accuracy compared to noise-free gradient updates that do not protect privacy in the strong sense afforded by differential privacy are essentially unavoidable. The addition of a privacy requirement inevitably constraints the learner's problem, and privacy must be traded-off with accuracy. Yet, the solution provided by Chen et al. (2022), while providing good performance, may still enjoy sub-optimal privacy-accuracy trade-offs. Can we develop new mechanisms with improved privacy-accuracy trade-off compared to the mechanism of Chen et al. (2022)?

Our starting point to address this question is to note that much research focusing on quantization in federated learning for the sake of communication complexity absent privacy (Alistarh et al., 2017; Reisizadeh et al., 2020; Haddadpour et al., 2021; Youn et al., 2022) reveal that performance degradation from quantization alone is somewhat minimal. Furthermore, quantization itself inherently reduces the amount of information encoded about the original input. While quantization in itself is insufficient for privacy, we posit that a two-stage approach, selecting a *randomized* quantization scheme followed by randomized rounding, can provide a viable approach to obtaining low communication complexity with formal *differential privacy* guarantees while still enjoying good performance. Thus:

> *Can we harness randomization in quantization schemes to further improve privacy-accuracy trade-offs in differentially private federated learning?*

To address this question, we introduce what we call the *Randomized Quantization Mechanism*, or *RQM* for short. RQM achieves privacy entirely through randomly sub-sampling quantization levels followed by a (randomized) rounding procedure to a close-by quantization level.

**Summary of contributions**  As mentioned above, our paper studies mechanisms for releasing gradients while satisfying Rényi differential privacy, and how our proposed mechanisms can be integrated in standard federated learning frameworks.

- In Section 5, we introduce our *Randomized Quantization Mechanism* that maps gradients to a randomized discrete grid in a way that preserves both standard and Rényi differential privacy.

- In Section 5.3, we provide theoretical evidence that our proposed Randomized Quantization Mechanism exhibits $\alpha-$Rényi differential privacy guarantees "locally", at the level of each single end-user device. Our theoretical guarantees hold for $\alpha \to +\infty$, implying in particular that they hold not just for Rényi but also for traditional $(\epsilon, 0)$-differential privacy. Also, we show that for any given $\alpha$, RQM provides lower Rényi divergence hence better Rényi DP guarantees than Poisson Binomial Mechanism (PBM) introduced by Chen et al. (2022) via the numerical Rényi DP computation approach. Further, in Section 5.4, we discuss the performance of RQM compared to PBM in terms of numerical privacy-accuracy trade-offs by using the mean-squared error (MSE) as an accuracy measure.

- In Section 6, we provide federated learning experiments that highlight the performance of our mechanism. In particular, we show that RQM outperforms the state-of-the-art PBM in terms of privacy-accuracy trade-offs. We demonstrate that incorporating RQM into the standard differentially private federated learning framework results in higher model accuracy compared to PBM, while using the same hyperparameters that led to improved numerical Rényi DP for RQM in Section 5.3. This indicates that the accuracy improvement is achieved without compromising privacy. To be specific, we show this by experimentally exploring possible values of the parameter $\theta$ used in PBM, and showing that for each value of $\theta$, there exists an instantiation of the parameters of our algorithm, RQM, that leads to better accuracy and privacy guarantees *simultaneously*.

## 2  Related work

Both communication complexity and privacy concerns have been driving forces behind the development of Federated Learning. Federated optimization often uses two types of privacy-preserving techniques hand-in-hand. One is secure multi-party computation, which protects the communication between local devices and the learner, preventing an attacker from intercepting messages sent between them (Bell et al., 2020; Bonawitz

et al., 2017). One is information-theoretic privacy guarantees such as differential privacy (Dwork et al., 2014) that prevent inference of any given single local device's data from observed summary outputs (such as local gradient updates or the learner's model itself). For example, McMahan et al. (2017b) and Geyer et al. (2017) add a calibrated amount of Gaussian noise to the average of clipped local device updates based on the FedAvg (McMahan et al., 2017a) algorithm.

In this paper, we focus on providing robust Rényi differential privacy guarantees in federated optimization while maintaining high communication efficiency and good accuracy. Previous methods have often used an approach based on quantization followed by the addition of discrete noise to achieve both differential privacy guarantees and low communication efficiency. Agarwal et al. (2018) introduces the first communication-efficient federated optimization algorithm with differential privacy by incorporating quantization with the binomial mechanism. Kairouz et al. (2021) and Agarwal et al. (2021) employ discrete Gaussian and Skellam mechanisms, respectively, in conjunction with quantization and secure aggregation for enhanced privacy. However, the above methods lead to biased estimation due to the necessity of modular clipping. To address this issue, Chen et al. (2022) and Chaudhuri et al. (2022) propose unbiased mechanisms with improved privacy-accuracy trade-offs. Chen et al. (2022) encodes local devices' gradients into a parameter of the binomial distribution, allowing their mechanism to generate a sample from this distribution without the need for additive discrete noise. In contrast, rather than using known privacy mechanisms, Chaudhuri et al. (2022) introduces the *Minimum Variance Unbiased mechanism* (MVU) to enhance the privacy-utility trade-off by solving an optimization problem designed to minimize the output variance of the mechanism while adhering to local differential privacy and unbiasedness constraints. Enhancing this model, Guo et al. (2023) propose a more scalable MVU mechanism with better privacy-utility trade-off, achieved through a new interpolation procedure in the numerical design process. Despite their progress in improving the privacy-utility trade-off, these methods do not fully exploit the privacy advantages offered by randomized quantization itself.

Our research is not the first to leverage compression techniques to achieve both communication efficiency and provable privacy benefits without incorporating additive discrete noise (Li et al., 2019; Gandikota et al., 2021). Li et al. (2019) assume a Gaussian input vector distribution for their sketching algorithms to ensure differential privacy guarantees, which might not be strictly necessary. Gandikota et al. (2021) ultimately first quantize the gradients updates, then randomize the quantized gradients via differential private mechanisms such as randomized response or Rappor (Erlingsson et al., 2014). However, and to the best of our knowledge, our Randomized Quantization Mechanism is the first investigation that exclusively utilizes randomization of the quantization itself to attain improved Rényi DP guarantees within Federated Learning frameworks.

## 3 Preliminaries

### 3.1 Differential privacy

The main privacy technique for our Randomized Quantization Mechanism is differential privacy, defined as below.

**Definition 3.1.** ((Approximate) Differential Privacy (Dwork et al., 2006)) For $\epsilon, \delta \geq 0$, a randomized mechanism $\mathcal{M} : \mathcal{D} \to \mathcal{R}$ satisfies $(\epsilon, \delta)$-differential privacy if for any neighbor dataset $D, D' \in \mathcal{D}$ differing by the addition or removal of a single user's records, it holds that

$$\Pr(\mathcal{M}(D) \in E) \leq e^\epsilon \cdot \Pr(\mathcal{M}(D') \in E) + \delta$$

for all events $E \subset \mathcal{R}$.

In this paper, we also consider a variant of standard differential privacy called *Rényi Differential Privacy* (or *Rényi DP*), introduced in the seminal work of Mironov (2017). We develop mechanisms that guarantee Rényi DP and by extension traditional DP. The use of Rényi DP allows for tight privacy accounting throughout the training iterations. Rényi differential privacy relies on first understanding the notion of *Rényi divergence*:

**Definition 3.2.** (Rényi Divergence (Rényi, 1961)) Let $P$ and $Q$ be probability distributions defined over $\mathcal{R}$. The Rényi divergence of order $\alpha > 1$ is defined as

$$D_\alpha(P||Q) := \frac{1}{\alpha - 1} \log \mathbb{E}_{x \sim Q}\Big[\Big(\frac{P(x)}{Q(x)}\Big)^\alpha\Big].$$

Then, Rényi differential privacy is defined as follows:

**Definition 3.3.** (Rényi Differential Privacy (Mironov, 2017)) A randomized mechanism $\mathcal{M} : \mathcal{D} \to \mathcal{R}$ satisfies $(\alpha, \epsilon)$-Rényi differential privacy if for any neighbor dataset $D, D' \in \mathcal{D}$ it holds that

$$D_\alpha(P_{\mathcal{M}(D)}||P_{\mathcal{M}(D')}) \leq \epsilon. \tag{1}$$

When $\alpha \to \infty$, $(\alpha, \epsilon)$-Rényi DP in fact recovers standard $(\epsilon, 0)$-DP. However, Rényi DP provides a finer-grained definition of privacy in that its guarantees can be tailored to the specific value of $\alpha$ and corresponding Rényi divergence that one considers. We now state a major property of the Rényi divergence that is useful to our theoretical analysis.

**Lemma 3.4.** *(Monotonicity) $D_\alpha$ is nondecreasing in $\alpha$. I.e., $D_\alpha(P||Q) \leq D_{\alpha'}(P||Q)$ for all $1 \leq \alpha \leq \alpha' \leq \infty$.*

### 3.2 User-level privacy

In the context of federated learning, we employ differential privacy to mask the contribution of any individual local device, making it challenging for a potential adversary to discern whether a local device's dataset was utilized in the training process. As such, we need to extend the traditional item-level definition of differential privacy (Definition 3.1) by redefining what we mean by neighboring datasets. In this context, two datasets are considered neighboring if one dataset can be created by changing any subset of data points of a single user from the other dataset. This user-level perspective is relatively standard and is the same as the one studied by McMahan et al. (2017b) and Levy et al. (2021).

**Definition 3.5.** (User-level DP (Levy et al., 2021)) For $\epsilon, \delta \geq 0$, a randomized mechanism $\mathcal{M} : \mathcal{D} \to \mathcal{R}$ satisfies $(\epsilon, \delta)$-user level DP if for any neighbor dataset $D, D' \in \mathcal{D}$ satisfying $\mathrm{d_{user}}(D, D') \leq 1$, it holds that

$$\Pr(\mathcal{M}(D) \in E) \leq e^\epsilon \cdot \Pr(\mathcal{M}(D') \in E) + \delta$$

for all events $E \subset \mathcal{R}$, where $\mathrm{d_{user}}$ is defined with $n$ users as

$$D = (D_1, \cdots, D_n), \text{ where } D_i = \{z_{i,1}, \cdots, z_{i,m_i}\} \to \mathrm{d_{user}}(D, D') := \sum_{i=1}^n \mathbb{1}\{D_i \neq D_i'\}$$

## 4 Model

We consider a federated learning set-up comprised of three types of entities: there are $n$ end-user devices, one secure aggregator called *SecAgg*, and one learner. The learner's goal is to learn a machine learning model, parameterized by a $f$-dimensional vector $w \in \mathbb{R}^f$, using the data on the devices through Stochastic Gradient Descent (SGD). However, the learner does not access the data from the devices directly, both for communication efficiency and privacy reasons. Rather, at each time step $t$:

1. Each end-user device $i$ computes a coordinate wise L-inf clipped gradient $g_t^i \in [-c, c]^f$ locally using the data on that device only. This gradient is then encoded into an integer $z_t^i$. This integer can be seen as the index of a discrete level in a discretization of the space of potential gradients.

2. The secure aggregator receives one message $z_t^i$ from each device $i$, which encodes information about the gradient $g_t^i$ computed by $i$. The aggregator aggregates them into a single message $z_t = \sum_i z_t^i$.

3. The server decodes $z_t$, computes the corresponding gradient $\hat{g}_t$, and takes a gradient step $w_{t+1} \leftarrow w_t - \eta\hat{g}_t$.

The traditional approach to federated learning releases gradients exactly; this approach is, however, i) inefficient from a communication complexity perspective and ii) vulnerable when it comes to privacy. We address i) by discretizing (or "quantizing") the space of possible values of the gradients to a grid of size $m$ per coordinate of the gradient; in turn, we require only $f \times \log m$ bits to represent a single update by a single device. Regarding ii), we note that it is well-understood that releasing exact gradients can lead to privacy violations in that the secure aggregator and the learner can recover information about device $i$'s dataset $D_i$ through the gradient itself. To address this issue, instead of releasing the gradient $g_t^i$ directly, device $i$ releases a noisy quantization level $z_t^i = RQM(g_t^i)$, where RQM is a Randomized Quantization Mechanism that must satisfy Rényi differential privacy. The entire setup is described formally in Algorithm 1.

What algorithm 1 describes is essentially the well-known, generic *Differentially Private Stochastic Gradient Descent* approach to federated learning (McMahan et al., 2017b). The focus and novelty of our work, however, come from the design of the private mechanism RQM itself. We propose our new mechanism in Section 5, characterize its privacy guarantees in terms of standard and Rényi differential privacy theoretically in Section 5.3 and empirically in the Numerical Privacy Guarantees part of Section 5.3, and study the accuracy of the resulting model when using privately-released gradient updates to train it in Section 6.

---

**Algorithm 1** Distributed DP-SGD with RQM

---

1: **Input:** $N$ local devices, each local device dataset $D_i \in \mathcal{D}$ $(i = 1, \cdots, N)$, coordinate wise L-inf clipping threshold $c$, RQM parameters $(\Delta, m, q)$, server learning rate $\eta$, initial vector $w_0$, loss function $f(w, D)$
2: **for** $t = 0, \cdots, T - 1$ **do**
3:     Server broadcasts $w_t$ to $n$ sampled local devices from total $N$ local devices;
4:     **for** each local device $i$ in parallel **do**
5:         $g_t^i \leftarrow \text{Clip}(\nabla f(w_t, D_i))$;
6:         $z_t^i \leftarrow RQM(g_t^i)$;
7:         send $z_t^i$ to the secure aggregator SecAgg.
8:     **end for**
9:     SecAgg outputs $z_t = \sum_{i=1}^{n} z_t^i$;
10:    server decodes $\hat{g}_t \leftarrow -(c + \Delta) + \frac{2z_t(c+\Delta)}{n(m-1)}$;
11:    server finds $w_{t+1} \leftarrow w_t - \eta \hat{g}_t$.
12: **end for**

---

# 5 The Randomized Quantization Mechanism

In this section, we introduce our main building block for privacy in federated learning. This building block provides a mechanism for privately releasing a scalar aggregate statistic of a single user's data in the form of a new algorithm called the *Randomized Quantization Mechanism* (RQM). We remark that when dealing with $f$-dimensional vectors instead, we apply our Randomized Quantization Mechanism independently to each vector coordinate.

We first formally present our RQM mechanism, outlined in Algorithm 2. Since our mechanism relies on a discrete probability distribution to choose the quantization, we show how this probability distribution over the quantizations translates into a probability distribution over outcomes of our mechanism on any given input $x$; this distribution over outcomes is crucial to characterize the level of privacy obtained by our mechanism. Then, we theoretically analyze the both standard and Rényi differential privacy guarantees of RQM by using this distribution over outcomes and empirically show that RQM achieves better numerical privacy guarantees than PBM. Finally, we demonstrate the superiority of RQM over PBM in terms of numerical privacy-accuracy trade-offs.

## 5.1 Randomized Quantization Mechanism

In this section, we assume that each user outputs a continuous scalar input $x$ computed from their data; this can be viewed as the simplest case of local updates.

Our RQM algorithm is then comprised of three key components: (1) enlarging the output range beyond the input range and setting up evenly spaced quantization bins, (2) sub-sampling realized quantization levels, and (3) performing a randomized rounding procedure on the *sub-sampled* (and only those) discrete levels to map a value $x$ to a quantization level. Each of these steps is crucial in ensuring the Rényi DP guarantees of the RQM, as we describe below. Formally, in each step, we perform the following operations:

1. We establish the output range of our mechanism by first augmenting the size of the input range. We do so by adding $\Delta$ to the upper bound $c$ and subtracting $\Delta$ from the lower bound $-c$ on the input data. This augmentation of the range is necessary for privacy: if we use the same range for the output, the quantization output for the maximum input ($x = c$) would always be $c$ subsequently, leaking a lot of information about $x$. After this, we establish $m$ initial, evenly spaced quantization levels ($B(0), B(1), \cdots, B(m-1)$) within this output range, which will be potential outputs of our mechanism.

2. Instead of using the entire set of quantization levels from step (1), we randomly sub-sample feasible quantization levels. We do so by including each discrete level for quantization with a carefully chosen probability $q$. The randomization of the quantization levels is necessary for privacy; otherwise, a value of $x$ would always map to the fixed set of two quantization levels deterministically. This immediately breaks differential privacy.

3. We perform quantization on the sub-sampled discrete levels (and these sub-sampled levels only), achieving both robust privacy and unbiased estimation. We identify the quantization bin that houses the input $x$ and perform randomized rounding on $x$ within this interval. The specific probabilities employed for randomized rounding can be reviewed in Algorithm 2.

The randomized quantization and rounding procedures described above are also illustrated later on in Figure 1a.

---

**Algorithm 2** Randomized Quantization Mechanism

---

1: **Input:** $c > 0, x \in [-c, c]$, extend the upper bound and lower bound by $\Delta$, the maximum number of quantization levels $m$, include a certain quantization level with probability $q$
2: Set $X^{\max}$: $X^{\max} = c + \Delta$, max and min value of quantization levels is respectively $X^{\max}, -X^{\max}$.
3: Quantization bins: $i = 0, 1, \cdots, m-1 \rightarrow B(i) = -X^{\max} + \frac{2iX^{\max}}{m-1}$.
4: sub-sample feasible quantization levels:
5:     Always include $B(0), B(m-1)$ & $i = 1, 2, \cdots, m-2 \rightarrow$ include $B(i)$ with probability $q$.
6:     sub-sampled indices of quantization levels $\rightarrow i_1(= 0), i_2, \cdots, i_l(= m-1)$
7: Quantization step:
8:     Find $i_{j^*}(i_1 \leq i_{j^*} \leq i_l)$ that satisfies $x \in [B(i_{j^*}), B(i_{j^*+1})]$.
9:     Do randomized rounding on $x$ in this interval.
10:

$$z = \begin{cases} i_{j^*+1}, & \text{with probability } \frac{x - B(i_{j^*})}{B(i_{j^*+1}) - B(i_{j^*})} \\ i_{j^*}, & \text{o/w} \end{cases}$$

11: **return** $z$

---

A major desirata of our algorithm, shared with the Poisson Binomial Mechanism, is that it is an unbiased estimator of $x$. Unbiasedness is a desirable property of SGD-based algorithms, as bias can cause issues with convergence, for example in traditional DP-SGD.

**Claim 5.1.** (Unbiasedness of RQM) Let $x \in [-c, c]$. Conditional on the realization of $i_{j^*}$, the quantization level at the output $z$ of Algorithm 2 is unbiased. I.e.,

$$\mathbb{E}\left[B(z)|i_{j^*}\right] = x,$$

where the expectation is taken over the randomization in the rounding step in line 10 of Algorithm 2.

The proof of Claim 5.1 is provided in Appx. A.1. Since $B(z)$ is unbiased conditional on the realization of $i_{j^*}$, it follows immediately that Algorithm 2 itself is also unbiased.

We now provide a quick remark on the additional flexibility of parameter choices offered by our framework, RQM, over the Poisson Binomial Mechanism, and which will be crucial when it comes to improving privacy-accuracy trade-offs:

**Remark 5.2.** The hyperparameters within our RQM algorithm offer enhanced flexibility, allowing for a more nuanced hyperparameter optimization when compared to PBM. RQM has in fact three hyperparameters $\Delta, q, m$, while PBM has two hyperparameters $\theta, m$ (See Algorithm 2 in Chen et al. (2022)). At a fixed number of discrete levels $m$, i.e. at a fixed level of communication complexity, this allows us to search over a bigger space of output distributions of quantization levels than Chen et al. (2022) through the choice of $(q, \Delta)$. In Section 6, we show that this leads to RQM achieving better privacy-accuracy trade-offs than PBM.

## 5.2 Resulting discrete distribution of outcomes

Given an input $x$ and parameters $m, q, \Delta$, we can compute the discrete probability distribution of outputs $Q(x)$ of RQM over the set of potential quantization levels $B(0), B(1), \cdots B(m-1)$. This discrete probability distribution is given in Lemma 5.3:

**Lemma 5.3.** *Let $m \in \mathbb{N}$, and $q \in (0,1)$ be parameters of Randomized Quantization Mechanism $Q$. Define evenly spaced $m$ quantization levels $B(0), \cdots, B(m-1)$ as in Algorithm 2. Let $j$ be the unique integer such that $x \in [B(j), B(j+1))$. The probability distribution of outcomes of the Randomized Quantization Mechanism is given by:*

$$
Pr\Big(Q(x) = i\Big) = \begin{cases} (1-q)^{j-i}\Big((1-q)^{m-j-2}\frac{B(m-1)-x}{B(m-1)-B(i)} + \sum\limits_{k=j+1}^{m-2} q(1-q)^{k-j-1}\frac{B(k)-x}{B(k)-B(i)}\Big), \ i = 0, \\ q(1-q)^{j-i}\Big((1-q)^{m-j-2}\frac{B(m-1)-x}{B(m-1)-B(i)} + \sum\limits_{k=j+1}^{m-2} q(1-q)^{k-j-1}\frac{B(k)-x}{B(k)-B(i)}\Big), \ 0 < i \leq j, \\ q(1-q)^{i-j-1}\Big((1-q)^{j}\frac{x-B(0)}{B(i)-B(0)} + \sum\limits_{k=1}^{j} q(1-q)^{j-k}\frac{x-B(k)}{B(i)-B(k)}\Big), \ j+1 \leq i < m-1, \\ (1-q)^{i-j-1}\Big((1-q)^{j}\frac{x-B(0)}{B(i)-B(0)} + \sum\limits_{k=1}^{j} q(1-q)^{j-k}\frac{x-B(k)}{B(i)-B(k)}\Big), \ i = m-1. \end{cases}
$$

$$(2)$$

The full proof of Lemma 5.3 is provided in Appx. A.2. Equation (2) exhibits different cases for $\Pr(Q(x) = i)$. These cases depend on i) how the $i$-th quantization level compares to $j$, where $j$ is defined to be such that $x \in [B(j), B(j+1))$ and ii) on the two special cases $i = 0$ or $i = m-1$. The probabilities corresponding to these two extreme values of $i$ differ from the rest in that we always incorporate the 0-th and $(m-1)$-th discrete level, which influences our probability calculations.

Figures 1a and 1b provide some insights into how to derive the distribution of outputs and gives some intuition for Lemma 5.3. In Figure 1a, the solid lines corresponding to the $(0, 3, 4, 7, 9, 10, 14, 15)$-th discrete levels have been selected for quantization, while the dotted lines $(1, 2, 5, 6, 8, 11, 12, 13)$-th discrete levels have been thrown away. To have $Q(x) = 10$, the 10-th discrete level must always be chosen while the 11-th discrete level must not be chosen by the sub-sampling step of our algorithm; the probability of this happening is $q(1-q)$. The probability of the next quantization level bigger than $x$ being the 14-th level, as shown in Figure 1a, is similarly given by $q(1-q)^2$ (levels 12 and 13 must not be sub-sampled, but 14 must be). Then, the likelihood of $x$ transitioning to the 10-th discrete level due to randomized rounding between the 10-th and 14-th levels is $\frac{B(14)-x}{B(14)-B(10)}$. I.e., the situation described in Figure 1a happens with probability $q^2(1-q)^3\frac{B(14)-x}{B(14)-B(10)}$. For a complete analysis, we must also account for randomized rounding intervals $[B(10), B(12)], [B(10), B(13)], [B(10), B(15)]$ and aggregate all these probabilities.

Figure 1b provides some insights on what the distribution induced by our quantization mechanism looks like, and seems to evidence that its shape differs from that of the Poisson Binomial Mechanism.

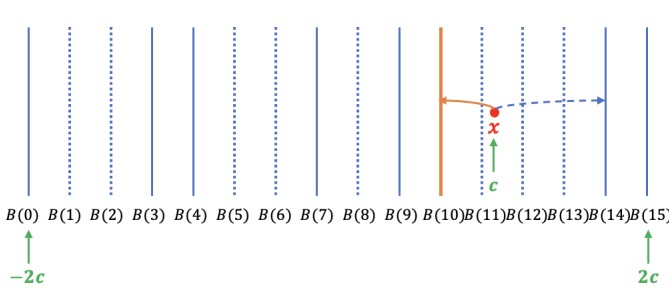 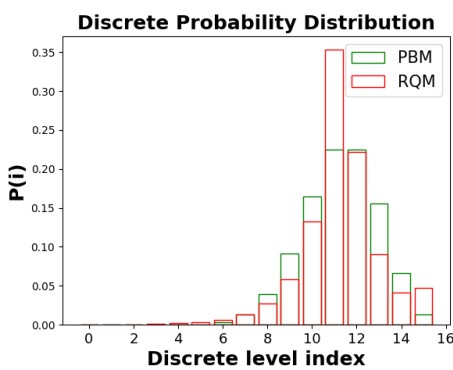

(a) An example of sub-sampling quantization levels for RQM.

(b) Distribution of outputs $Q(x)$ under PBM and RQM.

Figure 1: An example of RQM with input $x = c$ and parameters $\Delta = c$, $m = 16$.

### 5.3 Analysis of RQM's privacy guarantees

We now provide a theoretical analysis of the level of standard differential privacy achieved by our single-dimensional RQM mechanism. The full proof of Theorem 5.4 is provided in Appx. A.3.

**Theorem 5.4.** *(Standard DP) Let $c, \Delta > 0$, $m \in \mathbb{N}$, and $q \in (0,1)$ be parameters of Algorithm 2. Consider two scalars $x$ and $x'$ in $[-c, c]$, $P_{Q(x)}$ the distribution of outputs of RQM ran on scalar $x$, and $P_{Q(x')}$ the distribution of outputs of RQM ran on scalar $x'$. We have:*

$$D_\infty(P_{Q(x)}||P_{Q(x')}) \leq \log\left(2(1-q)^2\left(1+\frac{c}{\Delta}\right)\right) + m \log\frac{1}{1-q} = \epsilon. \tag{3}$$

*which indicates RQM is $(\epsilon, 0)$ differentially private.*

We show that RQM is $(\epsilon, 0)$ differentially private by leveraging the equivalence between $(\epsilon, 0)$-differential privacy and $(\infty, \epsilon)$-Rényi differential privacy. There, we note that the privacy level $\epsilon = \log\left(2(1-q)^2\left(1+\frac{c}{\Delta}\right)\right) + m \log\frac{1}{1-q}$ that we obtain increases linearly on $m$, the number of quantizations level. This makes sense as a large number of quantization levels allows one to encode more information about the initial scalar $x$, in turn leading to less privacy and higher $\epsilon$'s. We also note that as $\Delta$ increases, $\epsilon$ decreases, and we obtain more privacy; once again, this follows the intuition from Section 5.1 that when we increase the output range, we better protect the privacy of extreme values of $x$ that are close to $c$ or $-c$. As expected, when $\Delta = 0$, $\epsilon \to +\infty$ and our privacy guarantees are trivial, highlighting the fact that augmenting the range of output values beyond $[-c, c]$ is an unavoidable step to obtain reasonable privacy guarantees.

**Corollary 5.5.** *The upper bound in Theorem 5.4 immediately applies to Rényi differential privacy. For any $\alpha$, RQM is $(\alpha, \epsilon)$ Rényi differentially private:*

$$D_\alpha(P_{Q(x)}||P_{Q(x')}) \leq D_\infty(P_{Q(x)}||P_{Q(x')}) \leq \log\left(2(1-q)^2\left(1+\frac{c}{\Delta}\right)\right) + m \log\frac{1}{1-q} = \epsilon. \tag{4}$$

We obtain Corollary 5.5 by using Lemma 3.4.

**Remark 5.6.** In Corollary 5.5, we show the theoretical Rényi DP guarantees of RQM for the single dimensional case. For the multi dimensional case, the Rényi DP upper bound of Corollary 5.5 grows linearly in the dimension $f$ due to the composition theorems for Rényi DP (Mironov, 2017).

**Numerical Privacy Guarantees** In Theorem 5.4, we characterized the privacy guarantee of our Randomized Quantization Mechanism in the special case in which $\alpha \to +\infty$. We consider a *local* differential privacy

benchmark, evaluating privacy against a strong adversary that can see the output $Q(x_i)$ of *each* device $i$ (but not the input data $x_i$)[1]. To demonstrate the superior Rényi DP guarantees of RQM over PBM for arbitrary $\alpha$, we use equation 2 in Lemma 5.3 to numerically compute and plot the upper bound of Rényi divergence $D_\alpha(P_{Q(x)}||P_{Q(x')})$ for finite $\alpha$ in Figure 2. We compare it to the Rényi divergence of the Poisson Binomial Mechanism of Chen et al. (2022); we note that we do not compare to the upper bound provided by Chen et al. (2022) as the privacy measure that may not be tight, but instead to the actual Rényi divergence computed numerically and *exactly*. In both cases, we plot the nearly *worst-case* (over $x$, $x'$) Rényi divergence, which is approximately maximized when $x = c$ and $x' = -c$[2].

We fix the number of discrete levels $m$ as 16 for both RQM and PBM to compare privacy guarantees between the two algorithms *at equal communication complexity*. We set the value $c$ to be $1.5$[3]. We provide the numerical Rényi DP experiments comparing our results to PBM for a wide range of $\theta \in [0, 0.4]$ values considered by Chen et al. (2022). Specifically, we select three different $\theta$ values: 0.15, 0.25, and 0.35. For each $\theta$, the corresponding parameter pairs $(\Delta, q)$ for RQM are $(2.33c, 0.42)$, $(c, 0.42)$, and $(0.429c, 0.49)$, respectively. Figure 2 compares the Rényi divergence of PBM and RQM for a large range of $\alpha \in [0, 1000]$; we see significant disparities in the levels of Rényi privacy guaranteed by PBM and RQM for all different $\theta$s, with RQM vastly outperforming (i.e., guaranteeing a lower Rényi divergence hence a better privacy guarantee than) PBM, with the gap in privacy guarantees increasing as $\alpha \to +\infty$.

We have demonstrated that for typical ranges of parameters for PBM, we can find an implementation of our framework, RQM, that leads to significantly enhanced privacy. However, we note that getting better privacy on its own is easy (one could for example simply not use the data, and get perfect privacy at the cost of accuracy). In Section 5.4, we demonstrate that this improvement in privacy does not come at a cost in accuracy at the level of each single device, i.e. for $n = 1$. Then, in Section 6, we run larger scale federated learning experiments under multiple devices that further highlight that this improvement in privacy does not come at a cost in accuracy. Our experiments demonstrate that the same ranges of parameters also show accuracy improvements for our RQM method over PBM in a large-scale federated learning experiment on the EMNIST dataset.

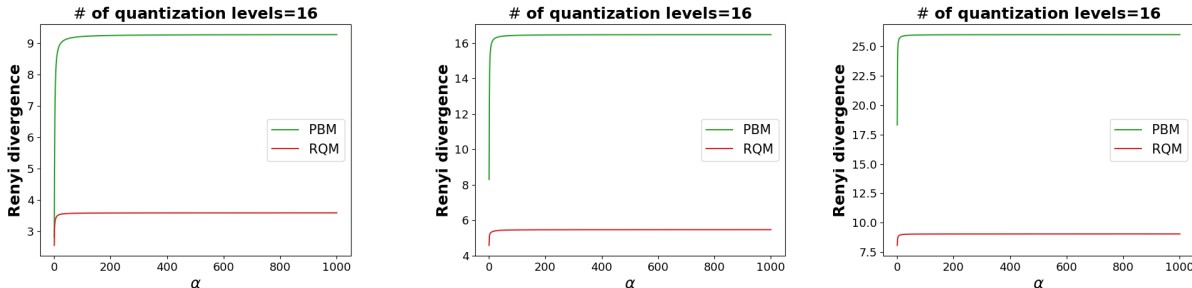

Figure 2: The results about Numerical Rényi privacy with $\theta = 0.15$ (left), $\theta = 0.25$ (middle), and $\theta = 0.35$ (right). All figures indicate how the Rényi divergence increases as $\alpha$ increases.

## 5.4 Privacy-Accuracy Trade-offs at the User Device Level

We present a numerical analysis of the privacy-accuracy trade-offs of our framework, highlighting the improved performance of RQM compared to PBM, in Figure 3.

To compute the privacy guarantee we achieve for a given set of parameters, we follow the numerical Rényi DP computation approach elaborated in Section 5.3.

---

[1]This is in contrast with central privacy, where the adversary can only see the aggregated gradients. The central privacy model requires trust in the aggregator, while the strong local privacy model does not.

[2]More details on this are discussed in Appx. B.1.1.

[3]Our mechanism is in fact scale-invariant for DP guarantees, and the choice of $c$ itself does not matter at a given constant ratio between $\Delta$ and $c$.

Our accuracy measure is the mean-squared error (MSE). The MSE is calculated as the average MSE over 30 equally spaced scalar input values $x$ within the range $[-c, c]$; the MSE for each value $x$ is based on $100,000$ samples to deal with privacy noise. In each plot in Figure 3, we fix $\alpha$ (= 1.5), $c$ (= 1.5), and $m$ (variable), ensuring that the same values are used for both PBM and RQM. Three privacy-MSE plots are shown in Figure 3, each corresponding to different values of $m$ (4, 16, and 64, mimicking Chen et al. (2022)).

Finally, to generate the privacy-MSE curves, we sweep through $\theta$ for PBM and $q$ for RQM, while keeping $\Delta = c$ constant for RQM; different values of $\theta$ for PBM and of $q$ for RQM correspond to different points on the curve. Across all plots in Figure 3, we consistently observe better privacy-MSE trade-offs with RQM compared to PBM. More numerical results with different $\alpha$ are provided in Appendix. B.1.3.

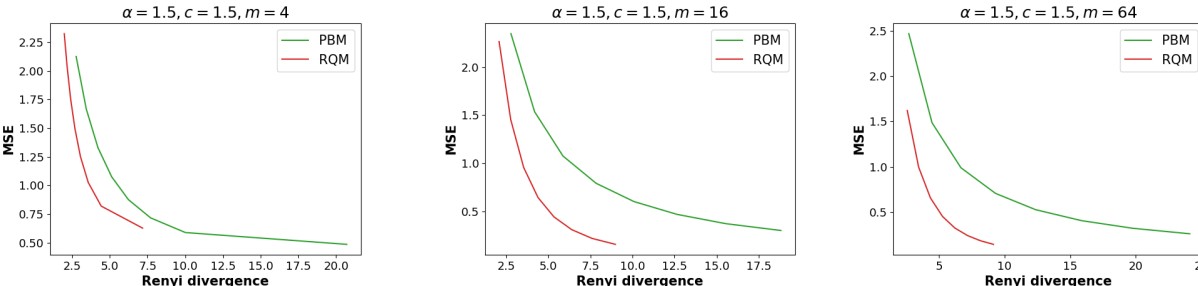

Figure 3: Numerical privacy-MSE trade-offs of RQM and PBM with $m = 4$ (left), $m = 16$ (middle), and $m = 64$ (right).

## 6 Federated Learning Experiments

In this section, we conduct experiments designed to complement our theoretical results and illustrate how RQM performs compared to PBM in terms of the privacy-accuracy trade-off on a larger scale Federated Learning experiment. We expand our experiments to provide insights beyond the privacy guarantees of the Randomized Quantization Mechanism itself, and to take into account how RQM integrates with the rest of the federated learning framework described in Section 4. We implement multi-dimensional RQM within the federated DP-SGD algorithm described in Algorithm 1. We employ the same parameters that yielded improved Rényi DP in the Numerical Privacy Guarantees part of Section 5.3 to demonstrate that RQM also excels in accuracy in our federated learning experiment.

We evaluate the privacy-accuracy trade-off of our algorithms against the current leading approach, the Poisson Binomial Mechanism (Chen et al., 2022), and an ideal noise-free clipped SGD benchmark that does not provide any differential privacy guarantee. The classification task for our federated learning experiment is performed on the EMNIST dataset (Cohen et al., 2017). The experimental setup details and more experimental results on the CIFAR-100 dataset are provided in Appx. B.2.

**Hyperparameter choice.** We adhere to the same hyperparameters for our FL experiments as those of the Numerical Privacy Guarantees part in Section 5.3: $m = 16$, $\theta = 0.15, 0.25, 0.35$ for PBM, $(\Delta, q) = (2.33c, 0.42), (c, 0.42), (0.429c, 0.49)$ for RQM. To highlight the flexibility of the choice of hyperparameters for RQM (Remark 5.2), we also plot results of two more $(\Delta, q)$ pairs for each $\theta$. For $\theta = 0.15$, we add two more pairs $(\Delta, q) = (4c, 0.5)$ and $(\Delta, q) = (c, 0.23)$; for $\theta = 0.25$, we add $(\Delta, q) = (2c, 0.57)$ and $(\Delta, q) = (0.66c, 0.33)$; and for $\theta = 0.35$, we include $(\Delta, q) = (c, 0.65)$ and $(\Delta, q) = (0.25c, 0.37)$. For clipping threshold $c$, we choose $2.9731 \times 10^{-5}$.

**Experimental results** The left and middle plots in Figure 4, where $\theta = 0.15$, clearly demonstrate that all three RQMs with different hyperparameter pairs show improved performance (in terms of loss and accuracy) on the EMNIST dataset than PBM. All three RQMs achieve similar accuracy. The performance of our three RQMs are still worse than noise-free clipped SGD: this is unavoidable because noise-free clipped SGD only focus on accuracy without providing any privacy guarantees, and is an ideal, impossible-to-achieve benchmark

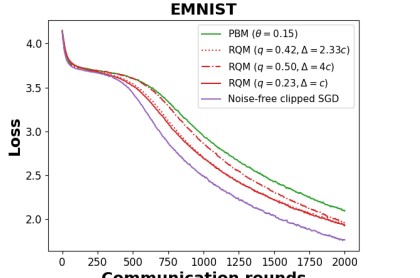 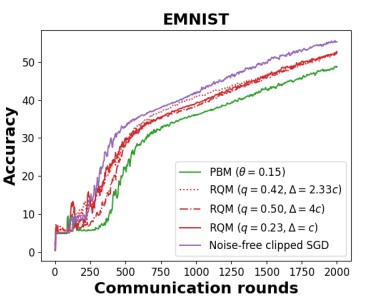 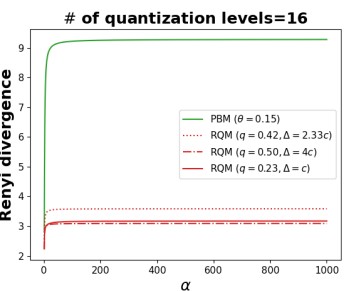

Figure 4: Comparing RQM with PBM ($\theta = 0.15$) and noise-free clipped SGD on EMNIST. All three RQMs with different hyperparameters outperform PBM in both a loss plot (Left) and an accuracy plot (Middle). These RQMs also show better Rényi DP guarantees than PBM (Right).

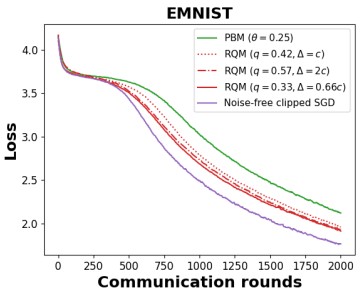 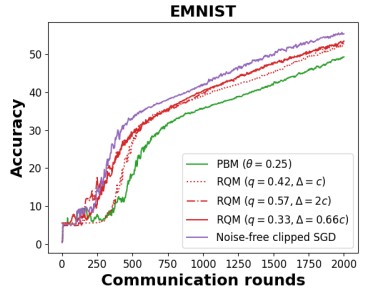 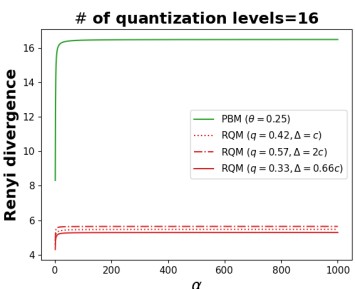

Figure 5: Comparing RQM with PBM and noise-free clipped SGD on EMNIST (Additional FL experiment with $\theta = 0.25$).

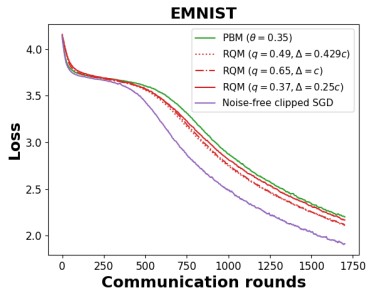 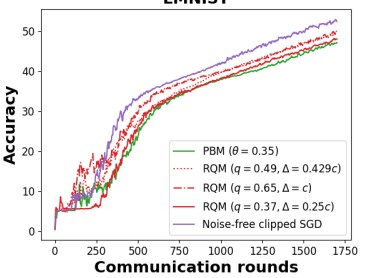 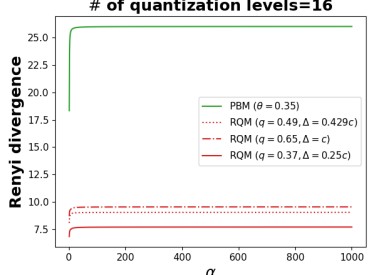

Figure 6: Comparing RQM with PBM and noise-free clipped SGD on EMNIST (Additional FL experiment with $\theta = 0.35$).

with privacy. Similarly, Figures 5 and 6, corresponding to $\theta$ values of 0.25 and 0.35 respectively, also show that all three RQMs outperform PBM in terms of performance. In Figure 5, among the three RQMs, the pair $(\Delta, q) = (0.66c, 0.33)$ achieves the highest accuracy. In Figure 6, the pair $(\Delta, q) = (c, 0.65)$ achieves the highest accuracy among the three RQMs.

The right plot in Figure 4[4] replicates experiment of the Numerical Privacy Guarantees part in Section 5.3 that were aimed at showcasing the privacy level achieved by RQM compared to PBM. The figure shows that the improved accuracy of the three RQMs compared to PBM in the left and middle figures does not come at the cost of privacy. In fact, the three plots together demonstrate that all three instantiations of RQM provide both better performance and better Rényi DP guarantees than PBM. I.e., in our experiments, RQM improves the *privacy-accuracy trade-off* of federated differentially private stochastic gradient descent compared to the current state of the art.

---

[4]The same conclusion can be drawn from Figures 5 and 6, but we highlight Figure 4 as a representative figure.

## 7 Discussion

In conclusion, this paper introduces a novel algorithm, the Randomized Quantization Mechanism (RQM). The RQM achieves privacy through a two-tiered process of randomization, which includes (1) the random subsampling of viable quantization levels, and (2) the application of a randomized rounding process with these subsampled discrete levels. We have theoretically demonstrated the Rényi differential privacy guarantees of RQM for a single end-user device and provided empirical evidence of its superior performance in the *privacy-accuracy trade-off* compared to the state-of-the-art Poisson Binomial Mechanism (PBM). In the future, it would be worthwhile to further examine the Rényi DP guarantees of RQM for multiple-device scenarios and the case of multi-dimensional RQM from a theoretical standpoint. Furthermore, increasing the flexibility of RQM hyperparameters by assigning unique probability values $q_i$ to each $i$-th discrete level presents an intriguing avenue for further enhancing the privacy-accuracy trade-off.

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

# A   Missing proofs

## A.1   Proof of Claim 5.1

We have that

$$
\begin{aligned}
&\mathbb{E}[B(z)|x \in [B(i_{j^*}), B(i_{j^*+1})] \\
&= B(i_{j^*+1}) \times \frac{x - B(i_{j^*})}{B(i_{j^*+1}) - B(i_{j^*})} + B(i_{j^*}) \times \frac{B(i_{j^*+1}) - x}{B(i_{j^*+1}) - B(i_{j^*})} \quad \text{(Line 10 of Algorithm 2)} \\
&= \frac{xB(i_{j^*+1}) - B(i_{j^*+1})B(i_{j^*}) + B(i_{j^*+1})B(i_{j^*}) - xB(i_{j^*})}{B(i_{j^*+1}) - B(i_{j^*})} \\
&= \frac{xB(i_{j^*+1}) - xB(i_{j^*})}{B(i_{j^*+1}) - B(i_{j^*})} \\
&= x.
\end{aligned}
$$

## A.2   Proof of Lemma 5.3

In Lemma 5.3, $j$ is defined as $x \in [B(j), B(j+1))$. We divide the range of $i$ into four cases-$0 < i \le j$, $i = 0$, $j + 1 \le i < m - 1$, $i = m - 1$- and compute the discrete probability $\Pr(Q(x) = i)$ for each case. The core proof idea of Lemma 5.3 is centered on evaluating the probability of each potential interval that can be used for randomized rounding for $x$. Subsequently, the probability that $Q(x) = i$ arises due to randomized rounding within a given interval is computed. Thus, when $i \le j$ and $k \ge j + 1$, we define the event $E_i$ and $F_k$ as below to use this notation for calculating the probability of each potential interval that can be used for the randomized rounding step in Algorithm 2.

$$
\begin{aligned}
E_i &: \text{ the event of } i\text{-th discrete level being used for randomized rounding} \\
F_k &: \text{ the event of } k\text{-th discrete level being used for randomized rounding} \quad (5)
\end{aligned}
$$

From the above definition of two events, $E_i \cap F_k$ indicates an event of the interval $[B(i), B(k)]$ being used for randomized rounding. In this event, this also means $i_{j^*} = i$ and $i_{j^*+1} = k$ in Algorithm 2. Now, let's deep dive into how we can exactly calculate $\Pr(Q(x) = i)$ for each case of four ranges.

(I) $0 < i \le j$:

First, Let us consider the case when $0 < i \le j$. Similar to the logic in Section 5.2, to have $Q(x) = i$, the $i$-th discrete level must always be chosen while the $(i + 1)$-th, $\cdots$, $j$-th discrete levels must not be chosen by the sub-sampling step of our algorithm. The probability of this happening is $q(1 - q)^{j-i}$. Thus, we can use the definition of the event $E_i$ in (5) for this case.

$$
\Pr(E_i) = \Pr(i: \text{ chosen}, (i+1, \cdots, j): \text{ not chosen}) = q(1 - q)^{j-i} \quad (6)
$$

Let us denote $k$ as an index of the next quantization level bigger than $x$. The possible $k$s are $j + 1, \cdots, m - 1$. When $k \in [j + 1, m - 2]$, the probability of the next quantization level bigger than $x$ being the $k$-th level is similarly given by $q(1 - q)^{k-j-1}$. Thus, we can use the definition of the event $F_k$ in (5) for this case.

$$
\Pr(F_k) = \Pr(k: \text{ chosen}, (j+1, \cdots, k-1): \text{ not chosen}) = q(1 - q)^{k-j-1} \quad (7)
$$

Then, the likelihood of $x$ transitioning to the $i$-th discrete level due to the randomized rounding between $i$-th and $k$-th levels is $\frac{B(k)-x}{B(k)-B(i)}$. This means

$$
\Pr(Q(x) = i|E_i \cap F_k) = \frac{B(k) - x}{B(k) - B(i)} \quad (8)
$$

Therefore, for $k \in [j+1, m-2]$, by combining (6), (7), (8), we get

$$
\begin{aligned}
&\Pr((Q(x) = i) \cap E_i \cap F_k) \\
&= \Pr(E_i \cap F_k) \cdot \Pr(Q(x) = i | E_i \cap F_k) \\
&= \Pr(E_i) \cdot \Pr(F_k) \cdot \Pr(Q(x) = i | E_i \cap F_k) \; (\because \; E_i, F_k : \text{ independent}) \\
&= q(1-q)^{j-i} \cdot q(1-q)^{k-j-1} \cdot \frac{B(k) - x}{B(k) - B(i)}
\end{aligned}
\tag{9}
$$

We can perform a similar computation for $k = m - 1$. However, the probability of event $F_{m-1}$ is different from that of Equation equation 7 because the $(m-1)$-th level is always chosen by Algorithm 2. Thus, we have

$$
\Pr(F_{m-1}) = \Pr(m-1 : \text{ chosen}, \; (j+1, \cdots, m-2) : \text{ not chosen}) = (1-q)^{m-j-2}
\tag{10}
$$

Therefore, for $k = m - 1$, by combining Equations (6), (10), and (8), we obtain

$$
\begin{aligned}
&\Pr((Q(x) = i) \cap E_i \cap F_{m-1}) \\
&= \Pr(E_i \cap F_{m-1}) \cdot \Pr(Q(x) = i | E_i \cap F_{m-1}) \\
&= \Pr(E_i) \cdot \Pr(F_{m-1}) \cdot \Pr(Q(x) = i | E_i \cap F_{m-1}) \\
&= q(1-q)^{j-i} \cdot (1-q)^{m-j-2} \cdot \frac{B(m-1) - x}{B(m-1) - B(i)}
\end{aligned}
\tag{11}
$$

Finally, by combining Equations (9) and (11), we get

$$
\begin{aligned}
&\Pr(Q(x) = i) \\
&= \sum_{k=j+1}^{m-1} \Pr((Q(x) = i) \cap E_i \cap F_k) \\
&= q(1-q)^{j-i} \left( (1-q)^{m-j-2} \frac{B(m-1) - x}{B(m-1) - B(i)} + \sum_{k=j+1}^{m-2} q(1-q)^{k-j-1} \frac{B(k) - x}{B(k) - B(i)} \right)
\end{aligned}
\tag{12}
$$

(II) $i = 0$:

We can compute $\Pr(Q(x) = 0)$ in a similar way as in case (I). However, the probability of event $E_0$ is different from that of Equation (6) because the 0-th level is always chosen by Algorithm 2. Thus, we have

$$
\Pr(E_0) = \Pr(0 : \text{ chosen}, \; (1, \cdots, j) : \text{ not chosen}) = (1-q)^j
\tag{13}
$$

Therefore, in (12), by substituting $E_i$ into $E_0$, we get

$$
\begin{aligned}
&\Pr(Q(x) = 0) \\
&= \sum_{k=j+1}^{m-1} \Pr((Q(x) = 0) \cap E_0 \cap F_k) \\
&= (1-q)^j \left( (1-q)^{m-j-2} \frac{B(m-1) - x}{B(m-1) - B(0)} + \sum_{k=j+1}^{m-2} q(1-q)^{k-j-1} \frac{B(k) - x}{B(k) - B(0)} \right)
\end{aligned}
\tag{14}
$$

(III) $j + 1 \leq i < m - 1$:

For $i$ within this range, we can similarly compute $\Pr(Q(x) = i)$ as in (I). To obtain $Q(x) = i$, the $i$-th discrete level must always be chosen while the $(j+1)$-th, $\cdots$, $(i-1)$-th discrete levels must not be chosen by the sub-sampling step of our algorithm. Thus, since $i \geq j + 1$, the probability of $i$-th discrete level being used for randomized rounding can be expressed by using $F_i$ (refer to (7)).

$$
\Pr(F_i) = \Pr(i : \text{ chosen}, \; (j+1, \cdots, i-1) : \text{ not chosen}) = q(1-q)^{i-j-1}
\tag{15}
$$

Let us denote $k$ as an index of the just previous level less than $x$. The possible $k$'s are $0, \cdots, j$. Then, for $k \in [1, j]$, the probability of the $k$-th discrete level being used for randomized rounding can be represented by utilizing $E_k$ (refer to (6)).

$$\Pr(E_k) = \Pr(k : \text{ chosen, } (k+1, \cdots, j) : \text{ not chosen}) = q(1-q)^{j-k} \tag{16}$$

Then, the likelihood of $x$ transitioning to the $i$-th discrete level due to the randomized rounding between $k$-th and $i$-th levels is $\frac{x - B(k)}{B(i) - B(k)}$. This means

$$\Pr(Q(x) = i | E_k \cap F_i) = \frac{x - B(k)}{B(i) - B(k)} \tag{17}$$

Therefore, for $k \in [1, j]$, by combining (15), (16), (17), we get

$$\begin{aligned}
&\Pr((Q(x) = i) \cap E_k \cap F_i) \\
=&\Pr(F_i) \cdot \Pr(E_k) \cdot \Pr(Q(x) = i | E_k \cap F_i) \\
=&q(1-q)^{i-j-1} \cdot q(1-q)^{j-k} \cdot \frac{x - B(k)}{B(i) - B(k)}
\end{aligned} \tag{18}$$

We can similarly calculate for $k = 0$ by using (13).

$$\begin{aligned}
&\Pr((Q(x) = i) \cap E_0 \cap F_i) \\
=&\Pr(F_i) \cdot \Pr(E_0) \cdot \Pr(Q(x) = i | E_0 \cap F_i) \\
=&q(1-q)^{i-j-1} \cdot (1-q)^j \cdot \frac{x - B(0)}{B(i) - B(0)}
\end{aligned} \tag{19}$$

Finally, by combining (18) and (19)

$$\begin{aligned}
&\Pr(Q(x) = i) \\
=&\sum_{k=0}^{j} \Pr((Q(x) = i) \cap E_k \cap F_i) \\
=&q(1-q)^{i-j-1} \left( (1-q)^j \frac{x - B(0)}{B(i) - B(0)} + \sum_{k=1}^{j} q(1-q)^{j-k} \frac{x - B(k)}{B(i) - B(k)} \right)
\end{aligned} \tag{20}$$

(IV) $i = m - 1$:

We can calculate $\Pr(Q(x) = m - 1)$ in a similar way compared to case (III). However, the $(m-1)$-th level should be always chosen by Algorithm 2, we rely on Equation (10) rather than Equation (15). We obtain:

$$\begin{aligned}
&\Pr(Q(x) = m - 1) \\
=&\sum_{k=0}^{j} \Pr((Q(x) = m - 1) \cap E_k \cap F_{m-1}) \\
=&(1-q)^{m-j-2} \left( (1-q)^j \frac{x - B(0)}{B(m-1) - B(0)} + \sum_{k=1}^{j} q(1-q)^{j-k} \frac{x - B(k)}{B(m-1) - B(k)} \right)
\end{aligned} \tag{21}$$

Therefore, we finally get Equation (2) of Lemma 5.3 from combining cases (I), (II), (III), and (IV).

### A.3 Proof of Theorem 5.4

Let's find an upper bound on $D_\infty(P_{Q(x)}||P_{Q(x')})$.

$$
\begin{aligned}
D_\infty(P_{Q(x)}||P_{Q(x')}) &= \sup_{i \in \{0,1,\cdots,m-1\}} \log\left(\frac{\Pr(Q(x)=i)}{\Pr(Q(x')=i)}\right) \\
&= \max\left(\sup_{i \in \{0,m-1\}} \log\left(\frac{\Pr(Q(x)=i)}{\Pr(Q(x')=i)}\right), \sup_{i \in \{1,\cdots,m-2\}} \log\left(\frac{\Pr(Q(x)=i)}{\Pr(Q(x')=i)}\right)\right) \\
&\leq \max\left(\log\left(\frac{1}{\min\limits_{i \in \{0,m-1\}} \Pr(Q(x')=i)}\right), \log\left(\frac{q}{\min\limits_{i \in \{1,\cdots,m-2\}} \Pr(Q(x')=i)}\right)\right)
\end{aligned}
$$

The second inequality comes from $\Pr(Q(x)=i) \leq 1$ for any $i$ and $\Pr(Q(x)=i) \leq q$ for $i \in \{1, 2, \cdots, m-2\}$. $\Pr(Q(x)=i)$ is less than or equal to $q$ for $i \in \{1, 2, \cdots, m-2\}$ because

$$
\begin{aligned}
\Pr(Q(x)=i) &= \Pr(Q(x)=i|i:\text{chosen})\Pr(i:\text{chosen}) \\
&= \Pr(Q(x)=i|i:\text{chosen}) \times q \\
&\leq q(\Pr(Q(x)=i|i:\text{chosen}) + \Pr(Q(x) \neq i|i:\text{chosen})) = q
\end{aligned}
$$

We establish a value for $j$ that makes it so that $-c$ falls within the range of values between $B(j)$ and $B(j+1)$. Since $\min\limits_{i \in \{0,m-1\}} \Pr(Q(x')=i) \geq \Pr(Q(-c)=m-1)$ and $\min\limits_{i \in \{1,\cdots,m-2\}} \Pr(Q(x')=i) \geq \Pr(Q(-c)=m-2)$, we obtain

$$
\begin{aligned}
D_\infty&(P_{Q(x)}||P_{Q(x')}) \\
&\leq \max\left(\log\left(\frac{1}{\min\limits_{i \in \{0,m-1\}} \Pr(Q(x')=i)}\right), \log\left(\frac{q}{\min\limits_{i \in \{1,\cdots,m-2\}} \Pr(Q(x')=i)}\right)\right) \\
&\leq \max\left(\log\left(\frac{1}{\Pr(Q(-c)=m-1)}\right), \log\left(\frac{q}{\Pr(Q(-c)=m-2)}\right)\right) \\
&= \max\left(\log\left(\frac{1}{(1-q)^{m-2} \cdot \frac{-c-B(0)}{B(m-1)-B(0)} + \sum\limits_{k=1}^{j} q(1-q)^{m-2-k} \cdot \frac{-c-B(k)}{B(m-1)-B(k)}}\right)\right. \\
&\quad \left., \log\left(\frac{q}{q\left((1-q)^{m-3} \frac{-c-B(0)}{B(m-2)-B(0)} + \sum\limits_{k=1}^{j} q(1-q)^{m-3-k} \frac{-c-B(k)}{B(m-2)-B(k)}\right)}\right)\right) \\
&\leq \log\left(\frac{1}{(1-q)^{m-2} \cdot \frac{-c-B(0)}{B(m-1)-B(0)}}\right) \\
&= \log \frac{1}{(1-q)^{m-2} \cdot \frac{\Delta}{2c+2\Delta}} \\
&= \log\left(\frac{2(1-q)^2(c+\Delta)}{\Delta}\right) + m\log\frac{1}{1-q}
\end{aligned}
$$

To go from the third to the fourth and fifth line, we used Lemma 5.3.

## B  More details about experiments in the main paper

### B.1  DP experiment

#### B.1.1  Nearly worst-case Rényi divergence

To numerically compute Rényi divergence, we use $\theta = 0.25$ for PBM and $(\Delta, q) = (c, 0.42)$ for our RQM. Under a single-device case, the peak Rényi divergence $D_\alpha(P_{Q(x_1)}||P_{Q(x'_1)})$ occurs predominantly around

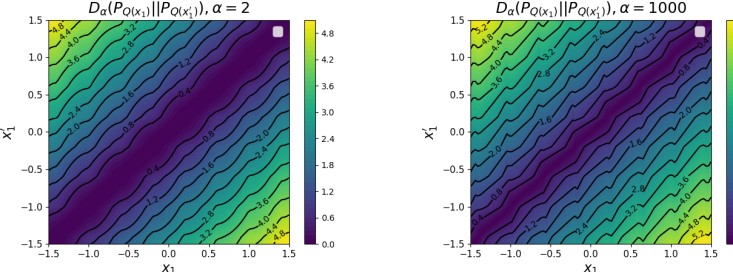

Figure 7: Both the left ($\alpha = 2$) and the right ($\alpha = 1000$) 2d plot illustrate how the Rényi divergence $D_\alpha(P_{Q(x_1)}||P_{Q(x_1')})$ changes with respect to the value of $x_1$ and $x_1'$ for the single-device scenario. Here, we follow the hyperparameter choice right above.

$(x_1, x_1') = (c, -c)$ and $(-c, c)$ (See Figure 7). Further, when we retain $x_1'$ at $-c$, as per Figure 8, it's discernible that Rényi divergence $D_\alpha(P_{Q(x_1)}||P_{Q(-c)})$ increases as $x_1$ transitions from $-c$ to $c$. In instances of larger $\alpha$, minor fluctuations at quantization levels are observed, followed by a swift incline in the Rényi divergence. However, considering these fluctuations as negligible, we deduce that the distance between distributions $P_{Q(x_1)}$ and $P_{Q(x_1')}$ rises almost monotonically as $x_1$ distances itself from $x_1'$. Furthermore, for a more quantitative analysis, the Rényi divergence computed at $(x_1, x_1') = (c, -c)$ is 5.46838, as shown in the right plot of Figure 8. At a quantization level near $x_1 = c$, where the Rényi divergence is locally maximized, the value is 5.46190, which is slightly less than the Rényi divergence at $x_1 = c$. Thus, in a single-device situation, we can judiciously choose $x_1 = c$ and $x_1' = -c$ to represent the scenario of worst-case Rényi divergence.

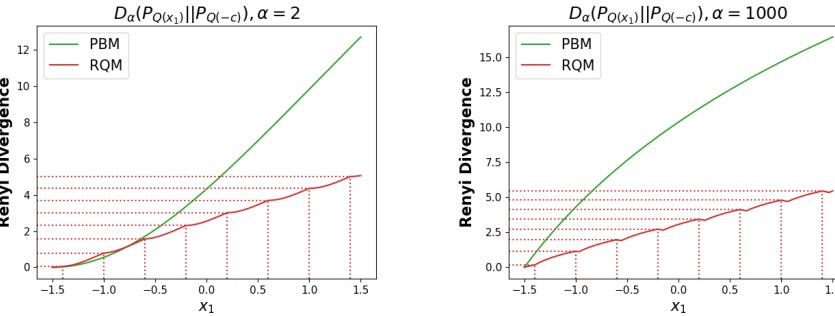

Figure 8: Both the left ($\alpha = 2$) and the right ($\alpha = 1000$) plot illustrate how the Rényi divergence $D_\alpha(P_{Q(x_1)}||P_{Q(-c)})$ changes as $x_1$ increases from $-c$ to $c$ for the single-device scenario. Here, we fix $x_1' = -c$.

### B.1.2   Zooming in on the leftmost plot of Figure 2

In Figure 9, we provide a closer examination of the leftmost plot from Figure 2, which highlights that RQM remains competitive in the very low $\alpha$ regimes. Additionally, RQM outperforms PBM for moderate $\alpha$ values, even before the convergence towards pure privacy is reached. Furthermore, the middle and right plots in Figure 2 reveal a distinct advantage of RQM over PBM, with a noticeable performance gap between the two.

### B.1.3   More results on privacy-MSE trade-offs

We conduct further numerical analysis on the privacy-MSE trade-offs in the lower $\alpha$ regime, where RQM and PBM exhibit similar Rényi divergence, as shown in Figure 9. In Figure 10, we fix $\alpha$ at 0.5, while keeping all other hyperparameters the same as those in Section 5.4. Even in the lower $\alpha$ regime, we observe that RQM outperforms PBM in terms of privacy-MSE trade-offs.

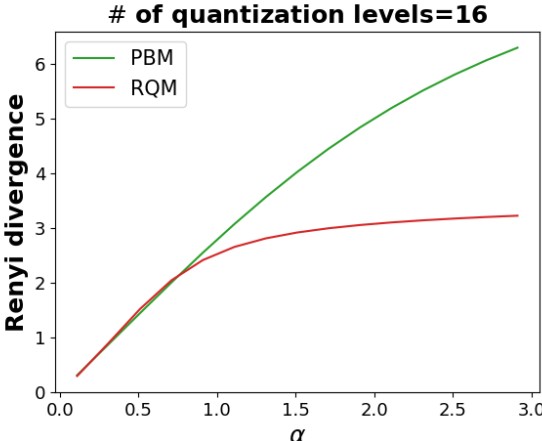

Figure 9: Comparison of the Rényi Divergence of RQM and PBM for the low regimes of $\alpha$.

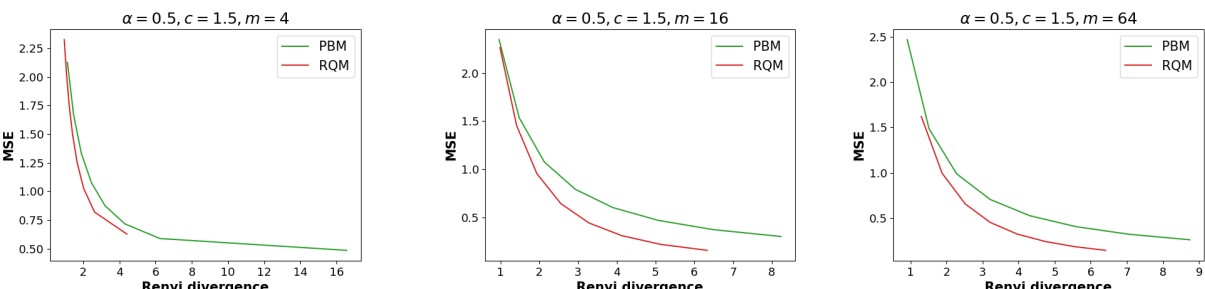

Figure 10: Additional numerical privacy-MSE trade-offs of RQM and PBM with $m = 4$ (left), $m = 16$ (middle), and $m = 64$ (right), when $\alpha$ is fixed as 0.5.

## B.2 FL experiment

**Implementation environment.** We adopt the same implementation setup as outlined in Chen et al. (2022). To implement our algorithm, we utilize TensorFlow (Abadi et al., 2015) and the TensorFlow Federated (TFF) library. Our computational resources include 2 NVIDIA RTX A5000 GPUs. We simulate a federated learning scenario involving a total of 3,400 local devices, with $n = 40$ local devices participating in each round. The total number of communication rounds is set to 2,000.

**Dataset & training model.** We perform image classification on the EMNIST dataset, which is comprised of 62 classes. We employ a Convolutional Neural Network (CNN) as the learning model for our training purposes.

### B.2.1 More FL experiment results on CIFAR-100

In Figure 11, we provide additional experimental results on CIFAR-100. We use $m = 16, \theta = 0.25$ for PBM and $\Delta = c, q = 0.42$ for RQM (same hyperparameters as those for EMNIST in Section 6), and clipping threshold $c = 5.3680 \times 10^{-5}$. We add the centralized continuous Gaussian mechanism and noise-free no-clipped SGD as the baselines. In Figure 11, we provide the evidence that our RQM achieves better performance in both loss and accuracy compared to the other methods including PBM, except for noise-free no-clipped SGD. Since we use the same hyperparameters for PBM and RQM as those in Figure 5 (dotted red line for RQM), the privacy level achieved by RQM is better than PBM. Thus, RQM shows improved privacy-accuracy trade-off compared to PBM for CIFAR-100 as well.

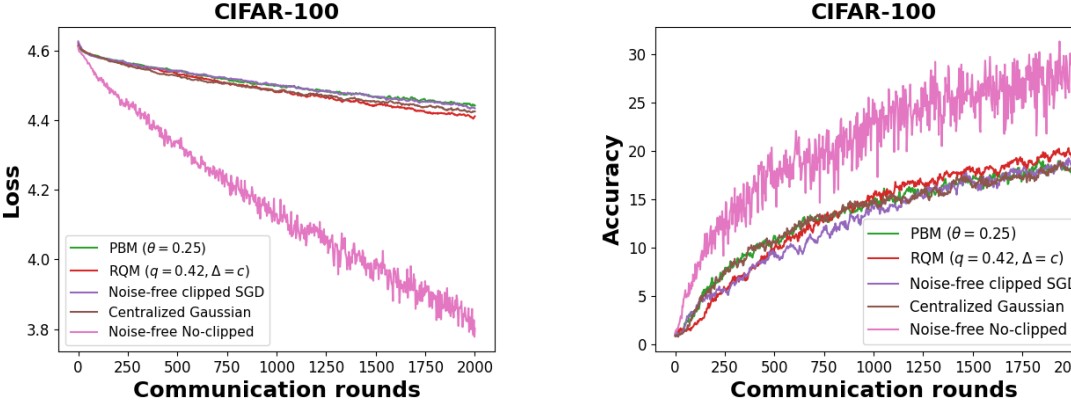

Figure 11: Comparing RQM with PBM, noise-free clipped SGD, centralized continuous Gaussian mechanism, and noise-free no-clipped SGD on CIFAR-100.

