# OpenReview forum: "Randomized Quantization is All You Need for Differential Privacy in Federated Learning"
_TMLR — Rejected by TMLR_

### Review · Reviewer_tYEG · 2024-08-14

**Summary Of Contributions:**

This paper proposes a new differentially private quantization mechanism called the Random Quantization Mechanism (RQM) to be used in federated learning.

This mechanism is described as follows: for a given input $x \in [-c,c]$, the interval $[-c - \Delta, c + \Delta]$ is regularly divided into $m$. Each quantization level is then kept with a fixed probability $q$. The returned value is drawn from the closest upper or lower kept quantization level.

The paper then derives RDP and pure DP guarantees for this mechanism. The proof relies on bounding the log ratio of probability by carefully deriving a closed form of the probability to choose each quantization level and then using simple inequalities.

Finally, the paper compares RQM to the Poisson Binomial Mechanism (PBM) for several parameter choices, using the EMNIST dataset.

**Audience:**

Yes

**Broader Impact Concerns:**

Sufficiently addressed.

**Claims And Evidence:**

No

**Requested Changes:**

I believe that authors should clarify the link with Laplace and provide a fairer comparison with PBM (i.e., explicit behaviors when $\alpha$ is not too big) before publication.

I suggest deriving, if possible, a similar theorem to Corollary 2.3 of PBM, and providing a heuristic to fix $\Delta$.

Please address the minor comments as well.

**Strengths And Weaknesses:**

**Strengths**
The paper focuses on aligning two constraints: privacy and communication efficiency, and proposes a method that improves both simultaneously. The paper is detailed and easy to follow. the code is clean and rather easy to follow.

**Weaknesses**
However, the presentation of the findings seems to obfuscate the underlying algorithm. Notably, RQM selects quantization levels with independent probability $q$. Up to the small terms coming from the extremities of the interval, the probability thus follows a geometric distribution of parameter $q$. This is very close to do quantization after applying the Laplace mechanism (as the geometric distribution is the discrete version of the exponential distribution). However, the authors never mention the Laplace mechanism. This is odd, especially as they compare RQM to PBM, which is explicitly inspired by quantization of Gaussian mechanism, and we clearly see the exponential decay of Laplace in figure 1 compared to the Gaussian bell curve.

In the experiments, one highlighted finding is that RQM has an advantage over PBM as $\alpha$ increases. This is a consequence of the well-known fact that the Gaussian mechanism is $(\epsilon, \delta)$-DP but not pure $\epsilon$-DP. In practice, many settings give an advantage to the Gaussian mechanism to avoid the heavy tail of Laplace, accepting approximate DP. Thus, the choice of $\alpha$ seems more motivated by artificially outperforming PBM than by enhancing privacy-utility trade-off.

Moreover, the authors do not compare RQM to the addition of discrete Laplace [1] (which I believe would be less costly to implement than their current implementation), nor to practical implementations of Laplace mechanisms [2] (where the encoding of floats could correspond to the level of quantization).

The authors praise the flexibility of RQM as one can tune $m$, $q$, $c$ and $\Delta$. However, in practice, avoiding extra parameters and knowing how to set them is more desirable. The experiments are not reassuring in this direction: for instance, $(\Delta, q) = (2.33c, 0.42)$ and $c = 2.9731 \times 10^{-5}$, making the comparison even more unfair towards PBM. A nice aspect of the PBM paper was the takeaway that excessive quantization is useless if the level of noise is high, as the variance of the final value is dominated by privacy randomness. Their corollary 2.3 allows choosing a relevant $m$ and its associated $\theta$. Here, there is no such result, which is a significant limitation to usability.

**Minor comments**:
- The choice of $f$ for the dimension is unusual on page 4.
- Typo "ENMIST dataset" on page 9.
- The proof at the end of page 15 can be simplified: it is a consequence of $A \subseteq B$ implies $P(A) \leq P(B)$.
- Although the code is provided, it could be beneficial to also report the information needed for reproducing experiments: architecture of the CNN used, details on how RQM was implemented (with a closed formula or with subsampling of quantization levels for each coordinate at each round? From the implementation I see that the close formula is implemented), and how the search for the interval was optimized, an how methods select the (unusual) hyperparameters finally used, number of runs. Including a readme with the package requirements could also help.

[1] A. Ghosh, T. Roughgarden, and M. Sundararajan. “Universally utility-maximizing
privacy mechanisms”. In: SIAM Journal on Computing 41.6 (2012), pp. 1673–
1693.

[2] Haney, Samuel et al. “Precision-based attacks and interval refining: how to break, then fix, differential privacy on finite computers.” ArXiv abs/2207.13793 (2022)

---

> ### Author Response · Authors · 2024-08-30
> **Rebuttal 1**
>
> Thank you for the helpful feedback. We hope to address your comments about the weaknesses of the paper.
>
> > W1: Comparison to quantization after applying the Laplace mechanism and comparison to discrete Laplace.
>
> There are several things we would like to draw attention to here. Please check the references in Rebuttal 2.
>
> First, we would like to ask and clarify whether the reviewer wants us to add comparisons to the discrete Laplace/geometric mechanism, the continuous Laplace mechanism that is then discretized as in [2, 3, 4], or both. We note that these two mechanisms are functionally different from each other.
>
> Second, we note that regardless of which discretized version of the Laplace mechanism we use, we note that there is a major difference between said mechanisms and the framework of the current paper: our mechanism uses a small, *finite* number of quantization levels, versus the mechanisms described in [1] and [2] are supported on the set of all integers. Of course, we can adapt [1] and [2] through clipping and rounding at no cost to privacy since those are simply post-processings. However, we note that this has significant implications for our mechanism:
> i) The first one is that the proposed discretized versions of Laplace lead to *biased* estimates of the underlying gradients that are then fed to our secure aggregator, while our quantization technique as well as that of PBM satisfy the additional constraint that they are unbiased. We acknowledge that we have not clearly stated this in the paper, and plan to do so in the next revision. [5] also mentions that the previous methods in [2, 3, 4] lead to biased gradient estimation issues. Bias issues are, in particular, why we chose PBM as a fair point of comparison.
> The reason we focus on gradient estimates that are unbiased is because those are known to work well with gradient-descent-based techniques for federated learning, and clipping is known to introduce convergence issues in DP-SGD types of algorithms.
> ii) The second one is that the "small terms coming from the extremities of the interval", are, in fact, not necessarily small, once again due to the clipping, and our mechanism can end up looking fairly different from the geometric mechanism and discretized versions of the Laplace mechanism.
> Mathematically, looking carefully at the probability distribution of RQM in Lemma 5.2, there is a $q(1-q)^n$ term that is effectively similar to the geometric distribution mentioned by the reviewer, but we note that the probability distribution of our RQM changes based on the location of input $x$ (or closest quantization level j) and how close it is to the clipping bounds $-c,~c$ (or quantization levels $i = 0$ and $i = m - 1$).
> Intuitively, if the value of $x$ is close to $0$ and in the middle of the $[-2c,2c]$ interval, it is indeed the case that we effectively recover a (truncated) version of the geometric mechanism. When $x$ becomes very close to $c$, for example, the distribution loses its symmetry, with tails dropping asymmetrically to the left and the right of $x = c$ (or quantization level 11 in Figure 1).
>
> > W2: "The choice of $\alpha$ seems motivated by artificially outperforming PBM."
>
> It is true that RQM has an advantage over PBM as $\alpha$ increases and we converge to pure $\epsilon$-DP as the reviewer noted. However, this does not mean that RQM is better than PBM only for large $\alpha$, and the choice of $\alpha$ is motivated by artificially outperforming PBM. We observe that the exact and numerical Renyi divergence of RQM are smaller than that of PBM for all values of $\alpha$.
>
> We acknowledge that this is currently hard to see on Figure 2 where most of the plot is in a regime where $\alpha$ is big enough that we effectively have converged to pure differential privacy. We zoom in on the leftmost plot of Figure 2 in the edited pdf (See Appx. B.1.2), where RQM remains competitive in the very low $\alpha$ regimes. Additionally, RQM outperforms PBM for moderate $\alpha$ values, even before the convergence towards pure privacy is reached. For the middle and the right plot of Figure 2, there is in fact a clear gap between RQM and PBM.

---

> ### Author Response · Authors · 2024-08-30
> **Rebuttal 2**
>
> > W3: Avoiding extra hyperparameters and knowing how to set them is more desirable.
>
> We agree with your argument that the small number of hyperparameters are more practical to use in many cases. However, in our case, RQM ($\delta, q, m, c$) only has one more hyperparameter compared to PBM ($\theta, m, c$), and the number of hyperparameters remains very small. Thus, as we mentioned in Remark 5.1, we believe that the positive effect of achieving better privacy-accuracy trade-offs from enhanced flexibility of RQM is bigger than the disadvantage of including a single additional hyperparameter.
>
> Further, since we only use 4 parameters (compared to modern ML models where tuning hundreds of parameters simultaneously is commonplace), hyperparameter optimization is simple in practice. While we do not provide theoretically-driven lessons on how to choose suitable parameters (our paper is primarily experimental), we note that even simple grid search is a viable and tractable option to implement in practice.
>
> Finally, we note that we have used the same constant $c$ for both RQM and PBM on our EMNIST experiments, and we are not fully sure that we understand the statement that our choice of c "makes the comparison even more unfair towards PBM". Could the reviewer please clarify?
>
> >W4: Minor comments
>
> We will address those in the next revision of the paper, in particular including the requested information to promote reproducibility of the code.
>
> [1] H. Zong, Q. Wang, X. Liu, Y. Li, and Y. Shao, "Communication reducing quantization for federated learning with local differential privacy mechanism", IEEE/CIC International Conference on Communications in China, 2021
>
> [2] N. Agarwal, A. Suresh, F. Yu, S. Kumar, and B. McMahan, "cpsgd: Communication-efficient and differentially-private distributed sgd", Advances in Neural Information Processing Systems, 2018
>
> [3] P. Kairouz, Z. Liu, and T. Steinke, "The distributed discrete gaussian mechanism for federated learning with secure aggregation", International Conference on Machine Learning, 2021
>
> [4] N. Agarwal, P. Kairouz, and Z. Liu, "The skellam mechanism for differentially private federated learning", Advances in Neural Information Processing Systems, 2021
>
> [5] W. Chen, A. Ozgur, and P. Kairouz. "The poisson binomial mechanism for unbiased federated learning with secure aggregation", International Conference on Machine Learning, 2022

---

> > ### Comment · Reviewer_tYEG · 2024-09-07
> >
> > I thank the authors for their responses.
> >
> > Regarding the comparison with existing Laplace mechanisms, I think both baselines would make sense. If I understand correctly, you claim the main difference is the unbiasedness. A naive question: would it be difficult to achieve unbiasedness for the discrete Laplace mechanism in a similar way? I cannot find where you prove this unbiasedness property (a search for "bias" yields 8 results, with the last one appearing at the beginning of page 6). I thank the authors for the discussion they provided about the behavior near the boundary, and I believe it would be beneficial to include it in the paper.
> >
> > Thanks for figure Appx. B.1.2. On the left, we see that PBM performs better. It would be helpful to discuss this point in more detail. I believe the fairness of the comparison is a shared concern, as highlighted in weakness 2 by reviewer mWDf, and I agree with the reviewer's questions.
> >
> > The fact that it was a small grid search is not a sufficient answer: even for machine learning model hyperparameters, we always try to reduce the number as much as possible. This is even more critical in a private setting, where tuning hyperparameters consumes additional privacy budget (did you take it into account?). In particular, in a real-world use case, it would be detrimental to have hyperparameters to tune in a decentralized setting with costly communication. Saying that there is only one more parameter to tune compared to PBM is misleading. In practice, $c$ is inferred from the problem, then $m$ and $\theta$ are optimized as described at the beginning of page 7 to minimize $m$ while respecting the privacy constraint. However, you do not provide clear instructions on how to choose $q$, $\delta$, and $m$, which makes a significant difference.
> >
> > Finally regarding your question: I said that the whole set of parameters $(\Delta, q)=(2.33 c, 0.42)$ and $c=2.9731 \times 10^{-5}$ on a single dataset sound a bit suspicious to establish empirically the superiority of RQM over PBM, I have nothing particular against this specific $c$, I guess that $3 \times 10^{-5}$ would perform similarly.

---

> > > ### Author Response · Authors · 2024-09-11
> > > **Further clarification 1 & please check our new revision pdf!**
> > >
> > > > "I cannot find where you prove this unbiasedness property  (a search for "bias" yields 8 results, with the last one appearing at  the beginning of page 6). I thank the authors for the discussion they  provided about the behavior near the boundary, and I believe it would be beneficial to include it in the paper."
> > >
> > > We did not include the claim in the paper and agree that it is an oversight on our end. This is now Claim 5.1 in the last revision of the paper.
> > >
> > > The result is actually a bit stronger than RQM being unbiased. In fact, conditional on the sampled quantization levels, the randomized rounding step itself is unbiased. The proof is very simple, and only relies on line 10 of Algorithm 2. For a given $x$ between realized quantization levels $i_{j^*}$ and $i_{j^*+1}$, we have that $B(z) = B(i_{j^*+1})$ with probability $\frac{x-B(i_{j^*})}{B(i_{j^*+1})-B(i_{j^*})}$, and $B(z) = B(i_{j^*})$ with probability $\frac{B(i_{j^*+1})-x}{B(i_{j^*+1})-B(i_{j^*})}$. Taking the average immediately shows that it is equal to $x$, providing unbiasedness.
> > >
> > > > "Regarding the comparison with existing Laplace mechanisms, I think both  baselines would make sense. If I understand correctly, you claim the main difference is the unbiasedness. A naive question: would it be difficult to achieve unbiasedness for the discrete Laplace mechanism in a similar way?"
> > >
> > > We believe this is an interesting question. We think this may be possible, and this may be another a good type of mechanism to explore in future work. We do not think this is quite trivial, however, and this would lead to a different looking mechanism, that is closer to ours (RQM).
> > >
> > > First, issues of truncating between $[-c,c]$ will also create asymmetries on how noise is added, with for example noise on values close to $-c, c$ being biased towards less extreme values after truncation (similarly to our discussion, in our original response, of the asymmetry of noise added close to the extremes.) The difficulty in debiasing will come from the fact that we will need to add different levels of noise for different values of $x \in [-c,c]$. It may be hard to call such a mechanism "Laplace" anymore at this point, given that it works fundamentally differently and cannot simply be explained by additive noise that is independent of the data $x$--we would need to end up doing something closer to our algorithm, RQM.
> > >
> > > Second, let us take the example of the continuous Laplace mechanism, then discretized, and let us go through a simple way of debiasing this mechanism. Imagine $x + Z$ (after Laplace noise addition) is between two quantizations levels $i_1$ and $i_2$. To ensure unbiasedness, we need to randomize between quantiles $i_1$ and $i_2$ with a probability that actually depends on the realization of $x + Z$; the closer $x + Z$ is to quantile $i_1$, the higher the probability of rounding to $i_1$ should be. This will end up looking relatively different from than the standard continuous Laplace then discretization mechanism and, in fact, once again, more similar to what we have in line 10 of Algorithm 2.
> > >
> > > We do believe here that the devil in the details of all these algorithms---while they look similar at a high level, the subtle differences between all these algorithms do matter. In fact, we would risk that the reviewer could interpret RQM as the proposed modification of Laplace or a Laplace-inspired mechanism, to deal with unbiasedness and truncation issues---but highlight that this is a serious modification and that Laplace could not be applied directly here, if we really want an unbiased algorithm.
> > >
> > > > "Thanks for figure Appx. B.1.2. On the left, we see that PBM performs better. It would be helpful to discuss this point in more detail. I  believe the fairness of the comparison is a shared concern, as  highlighted in weakness 2 by reviewer mWDf, and I agree with the  reviewer's questions."
> > >
> > > See the revision (Section 5.4 and Figure 3) and our last response to reviewer mWDf that addresses this point! Also, in Appx. B.1.3, we show that RQM achieves better privacy-MSE trade-offs compared to PBM even in the lower $\alpha$ regime (we set $\alpha=0.5$).

---

> > > > ### Author Response · Authors · 2024-09-11
> > > > **Further clarification 2 & please check our new revision pdf!**
> > > >
> > > > > "The fact that it was a small grid search is not a sufficient answer:  even for machine learning model hyperparameters (...)"
> > > >
> > > > We highlight two things here:
> > > > * regarding the point about having to incur costly communication and privacy costs to choose hyperpameters: we believe we do not have to run an entire FL experiment to do so. What we can do instead is characterize the privacy-accuracy trade-off on the grid of hyperparameter, as we do as an intermediary steps of the new Figure 3 in response to reviewer mWDf.  The reason we can do so is because i) for a given $\alpha$, we can numerically (but exactly) compute the Renyi DP guarantee, just computing the Renyi divergence using the distribution given in Lemma 5.2 and ii) we can estimate the MSE/accuracy as proposed by reviewer mWDf and as done in our new Figure 3.
> > > > * Another observation here is that we may not even need to do a 4D grid search. If our goal is not optimality but simply to perform better than PBM, we note that fixing $\Delta = c$, the same $m$ as used in RQM, and only optimizing over the 1D parameter $q$ leads in to better results in our experiments. The guidance here could be to follow the parameter choices of PBM and optimize over $q$ only (in the "Hyperparameter choice" paragraph of section 6, we also choose different $\Delta$s to show the flexibility of the choice of hyperparameters for RQM. But, as we mentioned here, fixing $\Delta = c$ is enough).
> > > >
> > > > >”I said that the whole set of parameters $(\Delta, q) = (2.33c, 0.42)$ and $c=2.9731 \times 10^{-5}$ on a single dataset sound a bit suspicious to establish empirically the superiority of RQM over PBM”
> > > >
> > > > We provide more FL experiment results on CIFAR-100 in Appx. B.2.1 (See Figure 11). We use $m=16, \theta=0.25$ for PBM and $\Delta=c, q=0.42$ for RQM (same hyperparameters as those for EMNIST in Section 6), and clipping threshold $c=5.3680 \times 10^{-5}$. We add the centralized continuous Gaussian mechanism and noise-free no-clipped SGD as the baselines. In Figure 11, we provide the evidence that our RQM achieves better performance in both loss and accuracy compared to the other methods including PBM, except for noise-free no-clipped SGD. Since we use the same hyperparameters for PBM and RQM as those in Figure 5 (dotted red line for RQM), the privacy level achieved by RQM is better than PBM. Thus, RQM shows improved privacy-accuracy trade-off compared to PBM for CIFAR-100 as well.

---

### Review · Reviewer_mWDf · 2024-08-21

**Summary Of Contributions:**

- The paper proposes double randomization extension to the Poisson Binomial Mechanism [1], where both the quantization bins are randomly chosen and the rounding to the bins are randomized.
- The proposed algorithm works as follows: for every client vector,
    - first do per-coordinate (l-inf) clipping to $[-c, c]$
    - expand the range to $[-c - \Delta, c + \Delta]$
    - then pick $m$ discrete levels to round values onto; each bin from these discrete levels have probability $q$ to be kept
    - each coordinate is then randomly rounded to one of the nearest bins
    - the client vector is then sent to server for secure aggregation
- the contributions of the paper mainly lie in
    - the extension of PBM to also randomize what bins are used
    - working out the probability distribution for each coordinate
    - working out the corresponding Renyi DP guarantees of the proposed randomization scheme
    - providing an experimental comparison to the PBM mechanism

[1] The poisson binomial mechanism for unbiased federated learning with secure aggregation, ICML 2022.

**Audience:**

Yes

**Broader Impact Concerns:**

The paper is mostly theoretical and may not require a broader impact statement.

**Claims And Evidence:**

Yes

**Requested Changes:**

- see weakness 1: it is highly suggested that privacy-vs-utility plots (not just alpha vs renyi divergence) are added to the paper for completeness
- see weakness 2: similarly, a direct comparison against PBM in terms of the RDP bounds and a comparison between the effects of hyper parameters would be useful
- paragraph 2: "…biased estimation due to gradient clipping"
    - Nit: the bias comes from per-coordinate modular wrap-arounds, as opposed to "gradient clipping". The client can send arbitrary model updates that isn’t a gradient.

EDIT (Aug 30):
It seems that there are two parts in the proof in the appendix that may benefit from further clarification:
- first inequality on page 16: "Since $\min_{i \in \{0, m - 1\} } ...$ ", can the authors provide more clarification here?
- last inequality on page 16: $\leq \log \left(\frac{1}{(1-q)^{m-2} \cdot \frac{-c-B(0)}{B(m-1)-B(0)}}\right)$. I'm not quite following this inequality

**Strengths And Weaknesses:**

strength
- the proposed mechanism is pretty clean and simple
- the numerical experiments suggest that the proposed mechanism outperforms its main baseline (PBM), allowing the proposed method to be an easy-to-implement drop-in replacement
- the paper is generally well-written and easy-to-follow; e.g. Fig. 1 is helpful in illustrating the main idea

weaknesses
1. there are no privacy-utility tradeoff curves presented in the paper (e.g. epsilon of RDP vs MSE/accuracy). This makes Fig 3, 4, 5 hard to interpret — e.g. do the different of the first two columns have same privacy? if not, the higher utility (lower loss, higher acc) is not meaningful
2. there isn’t a direct comparison to PBM in terms of the RDP bounds; given the similarity to PBM, this should be provided to give readers a better intuition on the trade-offs of the proposed mechanism (e.g. a table comparing the effects of the hyperparameters)
3. nit: the word "clip" is used ambiguously throughout the paper (e.g. algorithm 1, intro). It could be L2 clip (to cap the norm of the model update vector), L-inf clip (per-coord), modular clipping (wrap-arounds due to SecAgg). Would be great if each mention of "clip" is explained explicitly.

---

> ### Author Response · Authors · 2024-09-02
> **Rebuttal**
>
> Thank you for the helpful feedback. We hope to address your comments about the weaknesses of the paper.
>
> >W1: There are no privacy-utility tradeoff curves presented in the paper (e.g. epsilon of RDP vs MSE/accuracy). Do the different algorithms of the first two columns have same privacy?
>
> The PBM paper [1] shows their privacy-MSE trade-off curves in Figure 1. They obtain an upper bound of Rényi divergence in Theorem 3.3 and use this for the curves. The bound of Theorem 3.3., however, may not always be tight, and the effective privacy-accuracy trade-offs may be different from the ones provided by the theoretical bounds.
>
> To address this, instead, we characterize *realized* privacy-accuracy trade-offs experimentally. We numerically compute the numerical Renyi divergence to compare the actual realized privacy guarantees of PBM and RQM across different sets of parameters, rather than loose upper bounds. We use the experimental accuracy of RQM and PBM from our FL EMNIST experiments (rather than using MSE) for an accuracy/utility comparison.
>
> To answer your question, we in fact do not directly provide a privacy-accuracy trade-off curve. Instead, we compare several sets of parameters for PBM and RQM. For each set of parameters, we show (on different figures) that RQM performs better than PBM both in terms of privacy, and in terms of accuracy, for these given parameters. More precisely, we note that RQM has better privacy (lower Rényi divergence) than PBM in the left and middle plots of Figures 3, 4, and 5. Then, we show that RQM achieves better utility over the same parameters in the rightmost plot of Figures 3, 4, and 5 as well. The statement here is that for the parameter choices we explore for PBM, we can find parameter choices for our framework, RQM, that has both better privacy and better accuracy, in particular implying that we have a better privacy-accuracy trade-off *empirically*.
>
> The reason we do not make a direct privacy-utility plot is that we need to run a large scale federated learning experiment for a 4-dimensional grid of parameters to experimentally find the ``Pareto'' frontier. While tractable in practice, this is hard given our limited access to computational power.
>
> >W2: There isn’t a direct comparison to PBM in terms of the RDP bounds.
>
> Could you elaborate the term "direct comparison to PBM in terms of the RDP bounds"? Do you mean the comparison of the theoretical upper bounds of RQM and PBM? As we mentioned in W1, we instead computed the actual Renyi divergence of both algorithms numerically and observed that RQM outperforms (i.e., guarantees a lower Renyi divergence hence a better privacy than) PBM without costs to accuracy (according to experiments on ENMIST). Thank you!
>
> >W3: The word "clip" is used ambiguously throughout the paper (e.g. algorithm 1, intro).
>
> The "clip" in intro means modular clipping as you mentioned in Requested Changes. The "clip" in Algorithm 1 means coordinate wise L-inf clip. We make the term "clip" clearer in the revision version of the paper.
>
> >Further clarification on proof - part1: first inequality on page 16
>
> As you can see from Figure 1, $\min\limits_{i \in \textbraceleft 0, m-1 \textbraceright}\text{Pr}(Q(x^\prime)=i)$ becomes smallest when $x^\prime (\in [-c, c])$ is far from the $i$th quantization level $B(i)$. Thus, this probability is minimized when $x^\prime=-c, i = m-1$ (due to the symmetry, the probability is also minimized at $x^\prime=c, i=0$). The same logic is applied for $\min\limits_{i \in \textbraceleft 1, \cdots, m-2 \textbraceright}\text{Pr}( Q(x^\prime)=i) \geq \text{Pr}(Q(-c)=m-2)$. We consider two ranges of $i$, $i \in \textbraceleft 0, m-1 \textbraceright$ and $i \in \textbraceleft 1, \cdots, m-2 \textbraceright$, because the probability keeps decreasing from the peak probability until $i=m-2$ and $\text{Pr}( Q(x^\prime)=m-1)$ becomes slightly larger than $\text{Pr}( Q(x^\prime)=m-2)$ (See Figure 1 (b)).
>
> >Further clarification on proof - part1: last inequality on page 16
>
> We need to show that both terms in max are smaller than $\log \bigg( \frac{1}{(1-q)^{m-2}\cdot\frac{-c-B(0)}{B(m-1)-B(0)}} \bigg)$. It is trivial that the first term is smaller than $\log \bigg( \frac{1}{(1-q)^{m-2}\cdot\frac{-c-B(0)}{B(m-1)-B(0)}} \bigg)$ because the first term contains $(1-q)^{m-2}\cdot\frac{-c-B(0)}{B(m-1)-B(0)}$ in its denominator in $\log$. To show that the second term is also smaller than $\log \bigg( \frac{1}{(1-q)^{m-2}\cdot\frac{-c-B(0)}{B(m-1)-B(0)}} \bigg)$, it is enough to show that $(1-q)^{m-2}\cdot\frac{-c-B(0)}{B(m-1)-B(0)} \leq (1-q)^{m-3}\frac{-c-B(0)}{B(m-2)-B(0)}$. We can easily show this because $(1-q)^{m-2} \leq (1-q)^{m-3}$ and $B(m-1) \geq B(m-2)$.
>
> [1] W. Chen, A. Ozgur, and P. Kairouz. "The poisson binomial mechanism for unbiased federated learning with secure aggregation", International Conference on Machine Learning, 2022

---

> > ### Comment · Reviewer_mWDf · 2024-09-07
> > **thank you for the responses; further clarification needed on privacy-utility comparison**
> >
> > ## W1
> >
> > > To address this, instead, we characterize realized privacy-accuracy trade-offs experimentally. We numerically compute the numerical Renyi divergence to compare the actual realized privacy guarantees of PBM and RQM across different sets of parameters, rather than loose upper bounds.
> >
> > I'm a little confused. If the contribution is theoretical, then there should be basic comparisons using standard accounting tools (to arrive at an $\varepsilon$, either RDP $(\alpha, \varepsilon)$ or converted to plain $(\varepsilon, \delta)$-DP), and on standard metrics (e.g. MSE of aggregating some random vectors).
> >
> > > To answer your question, we in fact do not directly provide a privacy-accuracy trade-off curve. ... The reason we do not make a direct privacy-utility plot is that we need to run a large scale federated learning experiment for a 4-dimensional grid of parameters to experimentally find the ``Pareto'' frontier. While tractable in practice, this is hard given our limited access to computational power.
> >
> > My apologies, but I have trouble understanding the difficulty of setting up a basic $\varepsilon$ vs MSE experiment of aggregating, say, a set of random vectors, producing a plot akin to Fig. 1 of PBM paper (https://arxiv.org/pdf/2207.09916).
> >
> > **The main claim of the submission *is* that RQM is better than PBM.** I find it odd that such a comparison akin to Fig. 1 of PBM is not provided in the paper. If, for some reason, under the illustration of Fig. 1 that *RQM* is equally as, or worse than, PBM, the submission should clarify why, on top of just justifying alternative visualizations (comparing Renyi divergence and FEMNIST experiments).
> >
> >
> > > We use the experimental accuracy of RQM and PBM from our FL EMNIST experiments (rather than using MSE) for an accuracy/utility comparison.
> >
> > While this is useful and necessary, *just* having FL EMNIST experiments isn't sufficient, since model training necessarily have more confounding factors (e.g. choice of models, LRs, dataset, and other hyperparams).
> >
> > ## W2
> >
> > Please see responses to W1 -- I was mainly referring to a plot akin to Fig. 1 of PBM paper (https://arxiv.org/pdf/2207.09916).

---

> > > ### Author Response · Authors · 2024-09-11
> > > **Further clarification & please check our new revision pdf!**
> > >
> > > We apologize for the confusion! We misunderstood what the reviewer requested and understood it as providing privacy-accuracy trade-offs in the ENMIST experiments.
> > >
> > > We provide the requested privacy-MSE trade-off in Figure 3 of the revised manuscript, including a comparison to PBM. In each plot, we fix a value of $\alpha$ (the Renyi-DP parameter), $c$ (the bound on the norm of $x$), and $m$ (the number of quantization levels); the values we use are the same across PBM and RQM in each Figure.
> > >
> > > Then, we separately sweep across the $\theta$ parameter in PBM, and the the parameter $q$ in RQM at fixed $\Delta=c$. The MSE is estimated averaging over i.i.d., uniformly at random values $x \in [-c,c]$ for each gradient. Then, each presented curves for PBM and RQM represents the MSE-privacy trade-off for a fixed $\alpha, c, m$ across PBM and RQM, and for a fixed $\Delta$ for RQM. We find consistent improvements through our method over PBM (at the cost of an additional parameter, $q$).
> > >
> > > Please let us know if there are more experiments you would like us to run (e.g., more values of $\alpha$, etc.)!

---

### Review · Reviewer_cUet · 2024-08-28

**Summary Of Contributions:**

This paper proposes a new differentially private federated learning algorithm based on randomized quantization. The proposed algorithm first discretizes each component of the model trained by each user by dividing the range of the gradient into different bins. Then, the algorithm reports the bin where the true value lies with the highest probability. The R'enyi DP guarantee of the proposed mechanism is provided for the 1-dimensional case. Experiments show that the proposed mechanism can enhance the privacy-utility trade-off.

**Audience:**

Yes

**Broader Impact Concerns:**

I have no additional broader impact concerns.

**Claims And Evidence:**

Yes

**Requested Changes:**

1. The theoretical RDP guarantee is proved only for the 1-dimensional case. However, for the multi-dimensional case, the proposed RQM is applied to each component. Since each component of the gradient is data-dependent, for the $f$-dimensional case, the total privacy budget (under RDP) should grow linearly as $f$ increases if we simply use the composition bound for RDP. If this understanding is correct (please correct me if I am mistaken), I suggest adding a discussion on this in the privacy guarantee section for the proposed RQM. In the experiments, is the numerical calculation of the Rényi divergence applied to $f$-dimensional vectors or to each component individually?

2. In Algorithm 2, it appears that after discretization, the user sends the bin in which the true value lies with the highest probability (please correct me if I am wrong). Related to DP, the randomized response (RR) mechanism also reports the true value for discrete variables with the highest probability. It would be helpful if the author could discuss the relationship and differences between the proposed RQM and the classic RR.

3. Several simple typos need to be corrected. For example, "Renyi" should be corrected to "Rényi". Additionally, several lowercase letters in the references should be capitalized, such as "sgd" to "SGD". Please polish the paper accordingly.

**Strengths And Weaknesses:**

Strengths: This paper proposes a new algorithm based on Randomized Quantization (RQ). In federated learning, RQ is widely used to reduce computational cost. The paper demonstrates that by modifying the RQ mechanism (RQM), RQ can also provide a Rényi Differential Privacy (DP) guarantee. Moreover, the experiments show that the proposed RQM can enhance the utility of privacy-preserving federated learning. Additionally, the paper is well-written and easy to follow.


Weaknesses:

Several details, especially the theoretical guarantee of the high-dimensional case, should be specified. Please see the requested changes.

---

> ### Author Response · Authors · 2024-09-04
> **Rebuttal**
>
> Thank you for the helpful feedback. We hope to address your comments about the Requested Changes of the paper.
>
> >Requested Changes 1-1: The theoretical RDP guarantee is proved only for the 1-dimensional case. However, for the multi-dimensional case, the proposed RQM is applied to each component. Since each component of the gradient is data-dependent, for the $f$-dimensional case, the total privacy budget (under RDP) should grow linearly as $f$ increases if we simply use the composition bound for RDP. If this understanding is correct (please correct me if I am mistaken), I suggest adding a discussion on this in the privacy guarantee section for the proposed RQM.
>
> Thank you for your suggestion. Your intuition on the multi-dimensional case is correct. We add a discussion about this point in the revised version of the paper (See Remark 5.5).
>
> > Requested Changes 1-2: In the experiments, is the numerical calculation of the Rényi divergence applied to $f$-dimensional vectors or to each component individually?
>
> To answer your question, in the experiments, the numerical calculation of the Rényi divergence is applied to each component individually. In our empirical privacy analysis, we can also extend the current result to the multi-dimensional case similar to the theoretical privacy guarantee section. As you mentioned, composition theorems for RDP suggest that $\epsilon(\alpha)$ for multi-dimensional gradients increases linearly in the dimension $f$.
>
> >Requested Changes 2: In Algorithm 2, it appears that after discretization, the user sends the bin in which the true value lies with the highest probability (please correct me if I am wrong). Related to DP, the randomized response (RR) mechanism also reports the true value for discrete variables with the highest probability. It would be helpful if the author could discuss the relationship and differences between the proposed RQM and the classic RR.
>
> We understand RR as typically operating in binary spaces, where RR has some probability $p < 1/2$ of flipping a 0 to a 1 and vice-versa. Our algorithm does something similar, but in a higher-dimensional space.
>
> Of course, one can extend randomized response to non-binary settings, that are instead finite. In this case, RR operates on discrete inputs (say {0,...,k} for simplicity) and can be defined through a probability matrix P where $P_{ij}$ is the probability of switching each data point with value i to having a different value j. The space to optimize over and number of hyperparemeters however quickly quadratically blows up, compared to our framework that only requires 4. In this RR mechanism, the binning step can introduce bias. [1] deals with this bias issue but requires extremely careful calibration of P.
>
> We do believe that the mechanism is functionally different in a way that we randomize first then pick the bin. Also, there is no requirement in fact, to the best of our knowledge, that "the randomized response (RR) mechanism also reports the true value for discrete variables with the highest probability". Even in binary space, choosing $p > 1/2$, the mechanism can get the answer wrong most often than not---however, we can correct for this knowing that $p > 1/2$; in fact we can account directly for the bias introduced by $p$. The same is true for the multi-dimensional mechanism, allowing in principle arbitrary switches between discrete values/bins, so long as the bias can be accounted and corrected for. There is an interesting connection like [1], as we can run RR then adjust the gradients to correct for bias to obtain unbiased estimates of the gradients but with very careful modification (as we explained in the previous paragraph). Unbiasedness is a requirement of both PBM and our algorithm RQM.
>
> We are happy to add further discussions of this, and also further points that the reviewer may have on RR vs RQM!
>
> >Requested Changes 3: Several simple typos need to be corrected.
>
> All typos you mentioned are corrected in the revision version of the paper.
>
> [1] K. Chaudhuri, C. Guo, and M. Rabbat, “Privacy-aware compression for federated data analysis”, Uncertainty in Artificial Intelligence, 2022

---

> > ### Comment · Reviewer_cUet · 2024-10-08
> > **Replies to Rebuttal**
> >
> > Thank you very much for the rebuttal, and I apologize for the delayed response.
> >
> > It appears that my concerns from the previous round have been addressed well. After reviewing the feedback from the other reviewers, I think it would be beneficial to reflect on the effects of data dimensionality on the utility of the proposed mechanism. Since there is no theoretical privacy-utility guarantee, it may be helpful to include some experimental results or discussions regarding the dimensionality limitations.

---

> > > ### Author Response · Authors · 2024-10-08
> > > **Further clarification & questions to Reviewer cUet**
> > >
> > > Thank you for the feedback! Note that we added the privacy and composition result you mentioned in Remark 5.6.
> > >
> > > Regarding ``some experimental results (...) regarding the dimensionality limitations'', note that the Federated Learning experiments of Section 6 and of the Appendix on EMNIST and CIFAR-100, all use multi-dimensional (the dimension is 28 x 28 for EMNIST and 32 x 32 for CIFAR) gradients, not just 1-dimensional ones. In these experiments, we see empirical improvements over previous work.
> > >
> > > Is this what the reviewer had in mind? Would the reviewer would like to also ask us a limitation/conclusion section mentioning the work is mostly empirical, privacy and accuracy guarantees are only for the 1-dimensional case, and better understanding our work theoretically in the f-dimensional case is an important avenue for future work?

---

> > > > ### Comment · Reviewer_cUet · 2024-10-09
> > > > **Replies to Further clarification**
> > > >
> > > > Thanks for the clarification; I don't have any additional questions at this stage.

---

> > > > > ### Author Response · Authors · 2024-10-09
> > > > > **Thank you!**
> > > > >
> > > > > Sounds good! Thank you very much for the helpful feedback so far.

---

### Author Response · Authors · 2024-10-02
**Further questions on our last response?**

Dear Reviewers,

We would like to confirm make sure your questions have been adequately addressed, and to ask if there are any further questions following our last response. We sincerely appreciate your time and effort in reviewing our paper and providing such valuable feedback.

Thank you.

---

### Decision · Action_Editor_Vbk2 · 2024-12-02

**Recommendation:** Reject

**Comment:**

With two out of three reviewers recommending acceptance while the third was more hesitant, I decided to read the paper in detail myself. I sincerely apologise for the delay caused by this.

As a result of my review, I find the proposed RQM to be interesting and potentially acceptable to TMLR, but this would require a fair comparison of the method with PBM and reporting its privacy properties according to standard practice.

In order to be acceptable, the authors would at minimum need to address the following concerns:

1. Fair comparison with PBM. The privacy guarantees of PBM contain a $\frac{1}{n}$ multiplier for $n$ clients. As far as I can tell, RQM has nothing comparable as the privacy analysis only considers a single client. In this sense, the use of SecAgg in Algorithm 1 is superfluous, as it provides no benefit for your privacy analysis - the privacy-utility would appear to be similar without it. This is a major limitation that should be discussed in the paper clearly. This difference also likely makes the EMNIST comparison unfair in terms of privacy-utility tradeoffs, as the privacy guarantees of the two algorithms are likely not comparable on the level of the whole model. To be fair, you would need to compare the methods at equal privacy while considering the $\frac{1}{n}$ amplification for PBM that any real end-user would make use of.

2. Presenting numerical privacy accounting results transparently. While I very much support using numerical methods for computing privacy parameters, doing this properly requires a worst-case pair of datasets (or a dominating pair as defined by Zhu et al., AISTATS 2022). In the absence of a strict worst case, a proper way to express this would be for example to follow the example of Chua et al.: "How Private are DP-SGD Implementations?" (ICML 2024), and explicitly mark the numerical bound computed for one pair of datasets as a lower bound. Presenting numerical results for non-provably-worst-case without being explicit about this at every instance (every figure, table etc.) is misleading and unacceptable.

3. Choice of PBM parameters. PBM paper would appear to restrict $\theta \in [0, 1/4]$. You would need to justify your use of $\theta=0.35$ or drop it.

Additionally, I would highly recommend the authors to address the following concerns:

4. Clearly communicate the privacy-utility-communication trade-off of RQM in 1D in comparison with PBM. Fig. 3 is great and I would recommend starting the results with that and dropping Fig. 2, which is meaningless without somehow accounting for utility differences.

5. Extend the privacy-MSE comparison in Fig. 3 to smaller values of $\epsilon$ that would provide non-trivial formal protection.

6. Reporting the accuracy of RQM as a function of hyperparameters, similar to theory in the PBM paper.

7. Justifying the choice of hyperparameters used in the numerical examples.

Based on these considerations, it is clear that the paper is not acceptable for publication in its present form, and the required revisions are significant enough to warrant another round of reviewing. Therefore I recommend rejection with an option to resubmit a major revision.

**Audience:**

Yes, the paper would be interesting to the DP federated learning community.

**Claims And Evidence:**

No, the claims made in the submission are not supported by accurate evidence. The paper contains misleading differential privacy claims and it misrepresents the main baseline PBM.

**Resubmission Of Major Revision:**

The authors may consider submitting a major revision at a later time.